# Bri2 BRICHOS client specificity and chaperone activity are governed by assembly state

Gefei Chen[1], Axel Abelein[1], Harriet E. Nilsson [2], Axel Leppert[1], Yuniesky Andrade-Talavera[3], Simone Tambaro[1], Lovisa Hemmingsson[1,4], Firoz Roshan[3], Michael Landreh[5,6], Henrik Biverstål[1,7], Philip J.B. Koeck[2], Jenny Presto[1], Hans Hebert [2], André Fisahn[3] & Jan Johansson[1]

Protein misfolding and aggregation is increasingly being recognized as a cause of disease. In Alzheimer's disease the amyloid-β peptide (Aβ) misfolds into neurotoxic oligomers and assembles into amyloid fibrils. The Bri2 protein associated with Familial British and Danish dementias contains a BRICHOS domain, which reduces Aβ fibrillization as well as neurotoxicity in vitro and in a *Drosophila* model, but also rescues proteins from irreversible non-fibrillar aggregation. How these different activities are mediated is not known. Here we show that Bri2 BRICHOS monomers potently prevent neuronal network toxicity of Aβ, while dimers strongly suppress Aβ fibril formation. The dimers assemble into high-molecular-weight oligomers with an apparent two-fold symmetry, which are efficient inhibitors of non-fibrillar protein aggregation. These results indicate that Bri2 BRICHOS affects qualitatively different aspects of protein misfolding and toxicity via different quaternary structures, suggesting a means to generate molecular chaperone diversity.

[1] Department of Neurobiology, Care Sciences and Society, Center for Alzheimer Research, Division of Neurogeriatrics, Karolinska Institutet, 141 57 Huddinge, Sweden. [2] Department of Biosciences and Nutrition, Karolinska Institutet, and School of Technology and Health, KTH Royal institute of Technology, 141 83 Huddinge, Sweden. [3] Department of Neurobiology, Care Sciences and Society, Center for Alzheimer Research, Neuronal Oscillations Lab, Karolinska Institutet, 171 77 Stockholm, Sweden. [4] Department of Physics, Chemistry and Biology, Linköping University, 581 83 Linköping, Sweden. [5] Department of Chemistry, University of Oxford, South Parks Road, Oxford OX1 5QY, UK. [6] Science for Life Laboratory, Department of Microbiology, Tumour and Cell Biology, Karolinska Institutet, Tomtebodavägen 23 A, 171 65 Stockholm, Sweden. [7] Department of Physical Organic Chemistry, Latvian Institute of Organic Synthesis, Aizkraukles 21, Riga LV 1006, Latvia. Gefei Chen and Axel Abelein contributed equally to this work. Correspondence and requests for materials should be addressed to J.J. (email: janne.johansson@ki.se)

Molecular chaperones are essential for cellular homeostasis by promoting the correct folding of proteins over misfolding and aggregation. Under stress conditions, or when the natural protective mechanisms decline, proteins can misfold and self-assemble into large non-fibrillar amorphous aggregates as well as form fibrillar amyloid structures with toxic effects[1–3]. Non-fibrillar aggregates are associated with human diseases, e.g., cancer and cataract[4]. Self-assembly into amyloid fibrils of specific misfolded proteins is linked to about 40 human diseases, including the severe neurodegenerative disorders Alzheimer's disease (AD) and Parkinson disease, as well as common diseases like type-2 diabetes[5]. There are many different types and mechanisms of molecular chaperones preventing protein misfolding, ranging from large energy-dependent, multi-subunit complexes that are able to sequester and fold entire polypeptide chains in their interior to small chaperones that exert their function by attaching to exposed aggregation-prone regions of (partly) denatured polypeptide chains[6–8].

The ability to form amyloid fibrils is not limited to the proteins associated with disease, but a myriad of proteins has been shown to form amyloid fibrils in vitro, also generating protein species that are cytotoxic[2, 9]. The fact that only ~ 0.2% of all available human proteins have been found to form disease-associated amyloid in vivo suggests that efficient defense mechanisms that guard amyloidogenic peptides may exist. Molecular chaperones have been found to prevent amyloid fibril formation in vitro[10, 11] and the heat shock proteins Hsp70 and DNAJB6 were shown to suppress toxic effects of amyloid-forming polyglutamine repeat proteins in transgenic *Drosophila melanogaster* and mice[12, 13].

The most prevalent neurodegenerative disease is AD, for which formation of extracellular amyloid plaques and intracellular neurofibrillary tangles are histopathological hallmarks. AD development is associated with amyloid fibril formation of the 40–42 residue amyloid-β peptide (Aβ), in particular the more aggregation-prone Aβ42. Mutations in Aβ precursor protein (AβPP) or its processing enzymes result in early onset AD[14]. Exactly how dysregulated Aβ homeostasis translates into neurotoxicity and cognitive decline, however, remains to be established. Soluble, oligomeric forms of Aβ are found to be neurotoxic, apparently more than the mature fibrils, and may be key mediators of AD initiation and progression[15, 16]. Aβ42 fibril formation involves several nucleation steps that display different rate constants; oligomers are formed by primary nucleation from monomers, fibrils are extended by elongation, and oligomers are also formed on the surface of existing fibrils by secondary nucleation reactions[17]. The latter mechanism provides strong positive feedback, which dominates Aβ42 aggregation behavior, and generates the major part of cytotoxic oligomers[17]. Interference with discrete steps in the Aβ42 fibril formation pathways has different effects on the generation of toxic oligomers. Specifically preventing the secondary nucleation markedly reduces the amounts of oligomers, while blocking the fibril elongation event instead increases the amounts of toxic oligomers, although in both cases the overall fibril formation rate is reduced[18]. It is therefore important to delineate the exact mechanisms of fibrillization inhibitors in order to find ways to efficiently reduce Aβ42 neurotoxicity.

The BRICHOS domain (initially found in *Bri*2, *cho*n-dromodulin-1 and pro*s*urfactant protein C) is present in a set of proproteins that all have similar overall architecture and harbor regions that are prone to form β-sheets and to misfold into amyloid. BRICHOS has been proposed to assist the amyloid-prone region of their respective proprotein to fold correctly during biosynthesis[19, 20]. Bri2 is produced in several peripheral tissues and in the central nervous system (CNS), with significant expression in neurons of the hippocampus and cerebellum in humans[21, 22]. It affects AβPP processing[23] and deposits in AD amyloid plaques[24]. Mutations in Bri2 result in proteolytic release of amyloidogenic peptides (ABri and ADan) and eventually development of familial British or Danish dementias (FBD/FDD), which share clinical and pathological characteristics with AD[22, 25]. Recombinant human (rh) BRICHOS domains also affect fibril formation of amyloidogenic peptides that are not part of their precursors, including Aβ40 and Aβ42[26, 27]. Transgenic expression of the Bri2 BRICHOS domain alone, i.e., without the proprotein, inhibits Aβ42 fibril formation and toxicity in *Drosophila* CNS and eyes[28]. Recently it was shown that a mixture of Bri2 BRICHOS species with different quaternary structures affects the secondary nucleation and fibril elongation steps of Aβ42 fibrillization[29]. In addition, the mixture shows potent general molecular chaperone activity, measured as ability to prevent non-fibrillar aggregation of destabilized model substrate proteins[28]. The nature of the secondary nucleation sites is unknown and further studies of BRICHOS effects on Aβ42 fibrillization may provide valuable information on the way in which the secondary pathway functions[30].

Herein, we find that rh Bri2 BRICHOS can form well-structured particles with an apparent dihedral symmetry composed of 20–30 subunits, which efficiently inhibit non-fibrillar aggregation of thermo-denatured citrate synthase (CS).

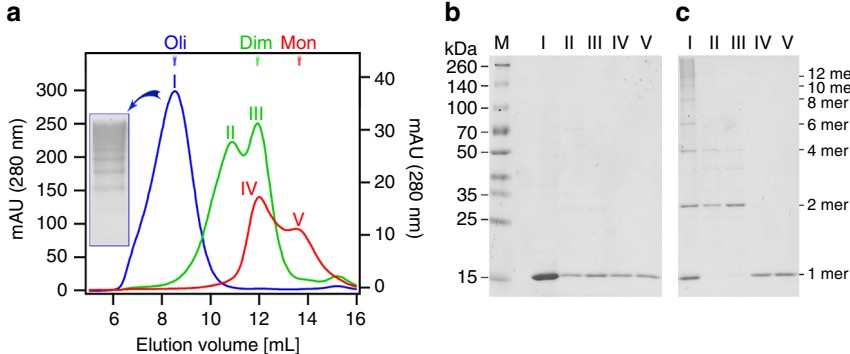

**Fig. 1** Quaternary structures of rh Bri2 BRICHOS. **a** SEC on Superdex 200 PG of rh Bri2 BRICHOS cleaved from individual rh NT*-Bri2 BRICHOS fusion protein species (see Supplementary Fig. 1). Oligomers (blue curve, absorbance scale on the left), dimers (green curve, absorbance scale on the right), and monomers (red curve, absorbance scale on the left). Inset shows native PAGE of the oligomer fraction. The different fractions indicated by Roman numerals in **a** were analyzed by SDS-PAGE under reducing **b** and non-reducing **c** conditions. Lane M in **b** shows migration of protein size markers with masses indicated to the left. The numbers of Bri2 BRICHOS monomer subunits likely present in the different species resolved in **c** are given to the right

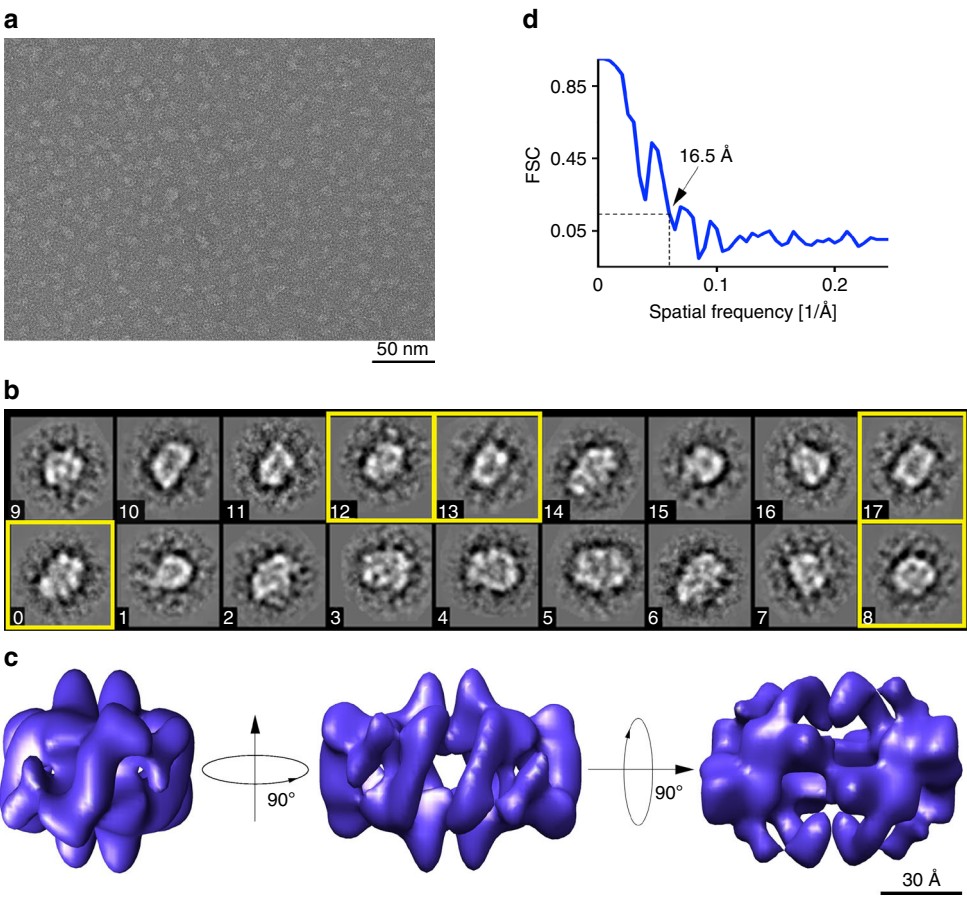

**Fig. 2** Electron microscopy analyses of Bri2 BRICHOS oligomers. **a** Transmission electron micrograph of negatively stained Bri2 BRICHOS oligomers recorded with a JEOL JEM2100F electron microscope and a DE-20 direct electron detector. **b** Representative 2D-classes of Bri2 oligomers. The class averages are consistent with an approximate 2-fold symmetry. The yellow frames indicate the most obvious 2-fold views. The side of each box side is approximately 22 nm. **c** 3D density map of Bri2 BRICHOS oligomer with dihedral (D2) symmetry. The viewing directions are parallel to the three different 2-fold axes. The map was based on 2718 particles extracted from images recorded on a DE-20 detector. The voxel size of the map is 2.076 Å. **d** The Fourier shell correlation (FSC) curve between reconstructions was produced by splitting the data set in two halves. Both halves were reconstructed separately. The resolution of ~17 Å for the reconstructed 3D density map was calculated from the curve at FSC = 0.143 (dotted lines)

Monomers and dimers, in contrast, are inactive against non-fibrillar aggregation but efficiently suppress Aβ42 toxicity in hippocampal slice preparations and amyloid fibril formation, respectively.

## Results

**Characterization of different rh Bri2 BRICHOS species**. To increase yields and allow isolation of distinct quaternary structure species, rh Bri2 BRICHOS was here produced with the new solubility tag NT*[31]. The fusion protein NT*-Bri2 BRICHOS could be resolved into oligomers, dimers and monomers by size exclusion chromatography (SEC) (Supplementary Fig. 1). Isolation of the NT*-Bri2 BRICHOS oligomers followed by proteolytic release of the BRICHOS domain gave Bri2 BRICHOS oligomers that migrate on SEC as a broad peak centered at an apparent molecular mass of about 370 kDa and show a polydisperse mixture of different sized oligomers on native PAGE (Fig. 1a, blue line, and inset). The oligomers contain an even number of disulfide-linked subunits, at least up to dodecamers, as judged from reducing and non-reducing SDS-PAGE (Fig. 1b, c, lane I). Isolation of NT*-Bri2 BRICHOS dimers followed by proteolysis resulted in two fractions upon SEC, with masses corresponding to dimers and tetramers, respectively, and both contain mainly disulfide-dependent dimers (Fig. 1, green line, lanes II and III).

Finally, release of Bri2 BRICHOS from the NT*-Bri2 BRICHOS fusion protein monomer gave monomers and dimers, both showing exclusively monomers on both reducing as well as non-reducing SDS-PAGE (Fig. 1, red line, lanes IV and V). The protein that eluted under the SEC peak V is monomeric according to electrospray ionization mass spectrometry (ESI-MS) (Supplementary Fig. 2a) and quantification of free thiols showed that the two cysteines are fully engaged in an intramolecular disulfide (Supplementary Table 1). The ESI-MS of the large oligomers shows a polydisperse appearance that is in line with the differently sized oligomers on native PAGE (Fig. 1a), and dimers were detached by collisional activation (Supplementary Fig. 2b, c), indicating that dimers are constituent building blocks of the oligomers. The monomers studied further herein were narrowly isolated by collecting fraction V in Fig. 1a, and the dimers and oligomers correspond to fraction III and I, respectively.

Circular dichroism (CD) spectroscopy showed the overall secondary structures are similar for all quaternary structure species, with a somewhat increased content of random coils in the monomer (Supplementary Fig. 2d). ProSP-C BRICHOS, for which there is a crystal structure determined[32], shows a similar CD spectrum as Bri2 BRICHOS species, albeit with a more pronounced minimum corresponding to random coil structure (Supplementary Fig. 2d), which is compatible with the fact that

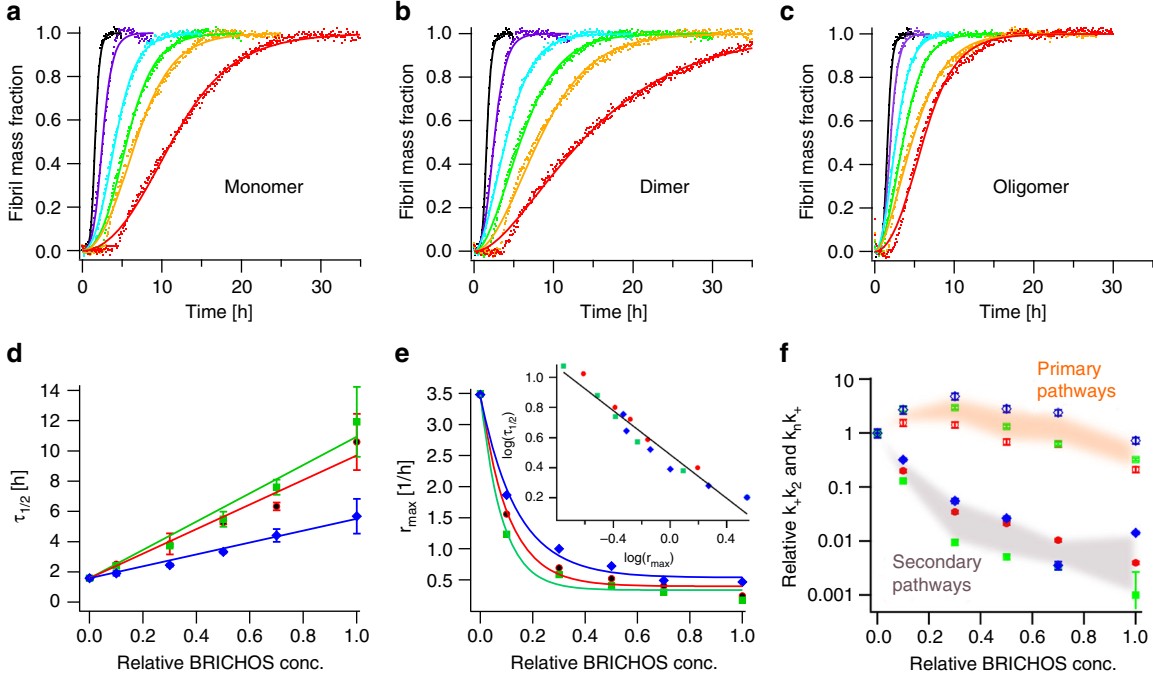

**Fig. 3** Kinetics analysis of rh Bri2 BRICHOS species effects on Aβ42 fibril formation. **a–c** Individual fits (solid lines) of normalized and averaged aggregation traces (dots) of 3 µM Aβ42 in the presence of 0 (black), 10 (violet), 30 (cyan), 50 (green), 70 (yellow) and 100% (red) Bri2 BRICHOS (molar percentage referred to monomeric subunits) with the combined rate constants $\sqrt{k_n k_+}$ and $\sqrt{k_+ k_2}$ as free fitting parameters. **d, e** The sigmoidal fitting parameter $\tau_{1/2}$ and $r_{max}$ exhibit a linear and mono-exponential dependence, respectively, on the relative Bri2 BRICHOS concentration for all species, i.e., the monomer (red), dimer (green) and oligomer (blue). Their logarithmic values fall on a line, indicating the same aggregation mechanism in the presence of all Bri2 BRICHOS species (inset graph). **f** The dependencies of the relative combined rate constants obtained from the fits in **a–c** reveal a strong effect of all Bri2 BRICHOS species on secondary nucleation ($k_+ k_2$, closed symbols and gray area), but not on primary ($k_n k_+$, open symbols and orange area) pathways. The data in (d-e) are presented as means ± standard deviations of 3–4 replicates of experiments that have been repeated at least five times with qualitatively similar results. The errors in **f** are presented as fitting errors

the loop between helices 1 and 2 is longer for proSP-C BRICHOS than for other BRICHOS domains[19]. One typical feature of molecular chaperones is binding to exposed hydrophobic patches of non-native polypeptides[33]. We therefore used the non-polar fluorescent dye bis-ANS for probing the presence of exposed hydrophobic areas of different rh Bri2 BRICHOS quaternary structures. Bis-ANS shows a blue shift of the emission maximum and increased emission intensity upon binding to exposed hydrophobic protein surfaces[34]. When incubated with bis-ANS, all rh Bri2 BRICHOS species gave a marked increase of emission intensity compared with bis-ANS in buffer, and also a blue shift of the emission maximum from about 533 nm to 480–490 nm (Supplementary Fig. 2e). Notably, the blue shift is smaller for dimers and monomers compared to the oligomers (Supplementary Fig. 2e). This indicates that all rh Bri2 BRICHOS species have exposed hydrophobic surfaces while the monomer and dimer apparently expose a different hydrophobic environment than the oligomer.

**3D reconstruction of rh Bri2 BRICHOS oligomers.** To obtain insights into the structural arrangement of rh Bri2 BRICHOS oligomers, we recorded transmission electron microscopy (TEM) micrographs from which 3D reconstructions can be calculated (Fig. 2a). The micrographs and 2D class averages revealed mostly homogenous assemblies (Fig. 2b). The biochemical analysis (Fig. 1 and Supplementary Fig. 2), showing that dimers (2 × 15 kDa) assemble into oligomers (~ 370 kDa), indicates the possibility of a 2-fold symmetry. Furthermore, the 2D classification suggests an approximate 2-fold symmetry, which was thus applied to the 3D reconstructions. We found that both dihedral

(D2) and cyclic (C2) symmetries gave a stable convergence in the 3D refinement (Supplementary Fig. 3). The refinement with applied C2 symmetry (one symmetry axis) finally converged to resemble the map refined with the higher D2 symmetry (three symmetry axes). Therefore, D2 symmetry was finally selected as shown in Fig. 2c. The resolution obtained in the final maps was ~17 Å with applied D2 symmetry (Fig. 2d), and 18 Å with C2 symmetry (Supplementary Fig. 3) according to the 0.143 FSC criterion. Indicated by the constructed 3D map, the rh Bri2 BRICHOS oligomers are highly structured assemblies, and the clefts and/or tori on the surface are potential hydrophobic binding sites (Fig. 2c), but still a higher resolution 3D map is necessary to elucidate the details. In order to evaluate roughly how many rh Bri2 BRICHOS subunits are able to be held in the constructed 3D map of the large oligomers, we calculated the volumes of the Bri2 BRICHOS monomer model built from proSP-C crystal structure (PDB 2YAD)[26], and the 3D map of the large oligomers, which are ~ 9340 Å³ and ~ 221,400 Å³ (with threshold set to 1), respectively. The volume comparison indicated that around 24 Bri2 BRICHOS subunits could be accommodated in the large oligomers, which is in good agreement with estimated masses of oligomers (~ 370 kDa) and monomers (15 kDa) by SEC.

**Aβ42 fibrillization affected by rh Bri2 BRICHOS species.** Shown by the thioflavin T (ThT) fluorescence assay, the rh Bri2 BRICHOS monomer, dimer and oligomer inhibit Aβ42 fibrillization in a dose-dependent manner (Fig. 3), and the aggregation kinetics follows a typical sigmoidal behavior. The appearances of the monomer, dimer and oligomers upon non-reducing SDS-

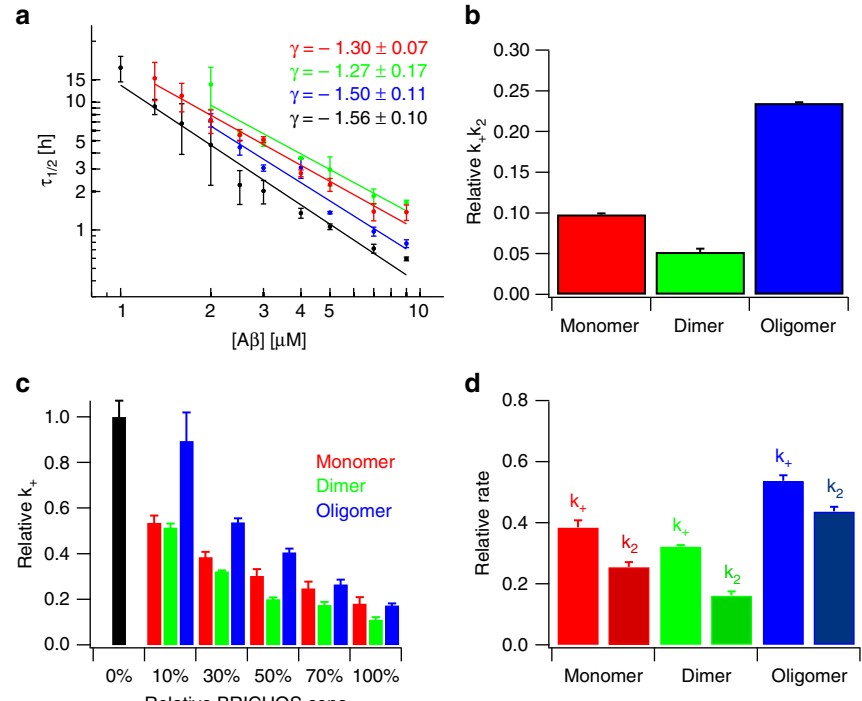

**Fig. 4** Effect of rh Bri2 BRICHOS on the nucleation mechanism of Aβ42. **a** Aβ42 in the absence (black) and presence of 0.9 μM Bri2 BRICHOS monomers (red), dimer (green) and oligomer (blue) exhibit a very similar dependence of the aggregation half time, $\tau_{1/2}$, on the initial peptide monomer concentration, described by the γ-exponent. **b** A global fit analysis from the data set in **a** revealed a dominate effect in $k_+k_2$, related to secondary nucleation and fibril-end elongation, where the Bri2 BRICHOS dimer is the most efficient species. **c** Estimation of the elongation rates from highly pre-seeded aggregation kinetics. **d** Estimated effect on the individual rate constants $k_+$ and $k_2$ from **b** and **c** by the different Bri2 BRICHOS species. The data in **a** and **c** are presented as means ± standard deviations of 3–6 replicates of experiments that have been repeated at least five times with qualitatively similar results. The errors of the γ-exponent in **a** and errors presented in **b** refer to the fitting errors. The errors in **d** are derived from the errors presented in **b** and **c**

PAGE are practically unchanged after 24 h incubations under the experimental conditions used for the ThT fluorescence assay (Supplementary Fig. 4). The aggregation half time, $\tau_{1/2}$, exhibits a linear relation on the Bri2 BRICHOS concentration, while the maximum rate of aggregation, $r_{max}$, shows a mono-exponential pattern (Fig. 3d, e). At equal mass concentration the rh Bri2 BRICHOS dimer, followed by the monomer, is the most effective structural species that prevents Aβ42 self-assembly. Notably, when correlating the sigmoidal fitting parameters in a double-logarithmic plot, the values of all rh Bri2 BRICHOS species fall on the same line, indicating that the inhibition mechanism is the same for all quaternary structural arrangements (Fig. 3e, inset).

In general, protein/peptide aggregation kinetics is determined by a set of microscopic nucleation events described by primary ($k_n$) and secondary ($k_2$ and $k_-$) nucleation rate constants in addition to fibril-end elongation ($k_+$)[35, 36]. Secondary nucleation can be dependent on the protein/peptide monomer concentration, e.g. the formation of a new nucleus on a fibril surface by $n_2$ monomers ($k_2$), or monomer-independent in the case of fibril fragmentation ($k_-$). The γ-exponent, defined by $\tau_{1/2} \alpha\ m(0)^\gamma$, is determined by the underlying nucleation mechanism[35, 37]. We examined the aggregation kinetics at different initial Aβ42 monomer concentrations, $m(0)$, in the presence of a constant rh Bri2 BRICHOS concentration, revealing roughly identical γ-exponents for all rh Bri2 BRICHOS species (Fig. 4a and Supplementary Fig. 5), which are in good agreement with previous results for Aβ42 aggregation alone under quiescent conditions[17]. This hence suggests that also in the presence of Bri2 BRICHOS Aβ42 fibril formation follows mainly monomer-dependent secondary pathways.

The set of kinetic equations determined by the microscopic rate constants can be solved analytically as shown by Knowles et al.[36–38] and fitting to the aggregation traces delivers two fitting parameters in form of the combined rate constants $\sqrt{k_nk_+}$ for primary and $\sqrt{k_+k_2}$ for secondary nucleation, respectively. To quantitatively elucidate the effects of Bri2 BRICHOS on Aβ42 aggregation, we first fitted the kinetic model individually to single aggregation traces of 3 μM Aβ42 in the presence of different Bri2 BRICHOS concentrations (Fig. 3a–c). This analysis revealed that primarily the fitting parameter $\sqrt{k_+k_2}$ associated to secondary nucleation is affected similarly by all Bri2 BRICHOS species (Fig. 3f). Subsequently, we recorded the aggregation kinetics at different Aβ42 concentration and constant Bri2 BRICHOS concentration. This set of aggregation traces can be fitted globally, where the fitting parameters $\sqrt{k_nk_+}$ and $\sqrt{k_+k_2}$ are constrained to the same value across all peptide concentrations. We found that the fitting parameter $\sqrt{k_nk_+}$ can be hold to the same value for all Bri2 BRICHOS species as for Aβ42 alone and the fits still describe well the kinetic data set (Supplementary Fig. 5 and Supplementary Table 2), supporting that mainly secondary pathways are modulated by the presence of Bri2 BRICHOS. The results summarized in Fig. 4b show that the Bri2 BRICHOS dimer is the most efficient species to inhibit nucleation events related to $k_+k_2$, i.e., fibril elongation or/and secondary nucleation.

To distinguish which of these nucleation events are predominantly affected, we performed aggregation kinetics in the presence of a high initial seed concentration. Under these conditions aggregation traces typically exhibit a concave aggregation behavior (Supplementary Fig. 6), where the initial slope is

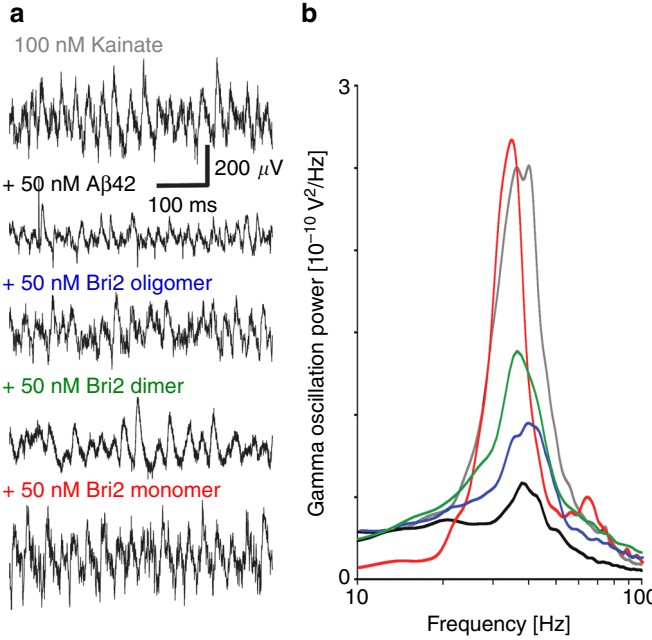

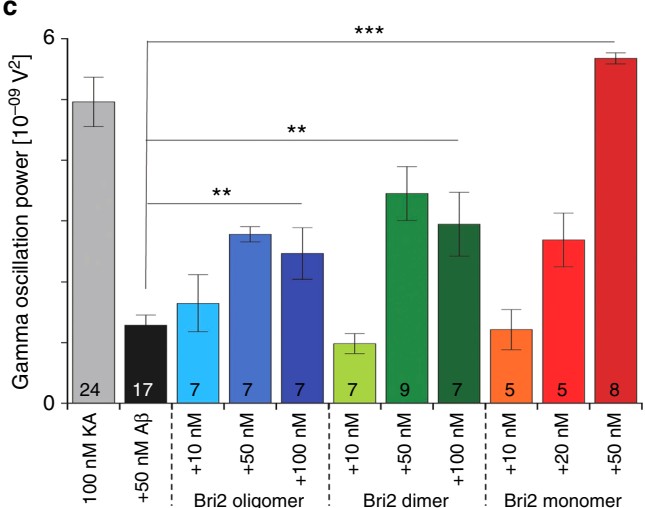

**Fig. 5** Effects on Aβ42 toxicity in mouse hippocampal slices of different rh Bri2 BRICHOS species. **a** Example traces of γ oscillations under control conditions (gray), after 15 min incubation with 50 nM Aβ42 (black) and after 15 min incubation with 50 nM Aβ42 + 50 nM oligomer, dimer or monomer (blue, green and red, respectively). **b** Example power spectra of γ oscillations under control conditions (gray), after 15 min incubation with 50 nM Aβ42 (black) and after 15 min incubation with 50 nM Aβ42 + 50 nM oligomer, dimer or monomer (blue, green and red, respectively). **c** Summary histogram of γ oscillation power under control conditions (gray), after 15 min incubation with 50 nM Aβ42 (black) and after 15 min incubation with 50 nM Aβ42 + three concentrations of oligomer, dimer or monomer (blue, green and red, respectively). The figures under the histograms denote number of biological replicates, and the data are reported as means ± standard errors of the means

directly proportional to the elongation rate $k_+$[35]. These experiments revealed that all Bri2 BRICHOS species decrease the elongation rate in a concentration-dependent manner (Fig. 4c and Supplementary Fig. 6). Remarkably, already at low Bri2 BRICHOS concentration, in particular the dimer and to less extent also the monomer, fibril-end elongation is noticeably retarded (Fig. 4c).

To check whether a sole inhibition of the elongation events can describe the observed aggregation behavior, we performed a global fit of the kinetic data set at constant Aβ42 and different Bri2 BRICHOS concentration where the fit was constrained such that only one single rate constant, i.e., $k_n$, $k_+$ or $k_2$, is the sole fitting parameter (Supplementary Fig. 7 and Supplementary Table 3)[39, 40]. As expected from the previous analysis, the primary nucleation rate $k_n$ as the sole fitting parameter gave only rise to insufficient fits, underlining that mainly secondary pathways are modulated. However, also a strict constraint where either $k_+$ or $k_2$ is the sole free fitting parameter does not describe the aggregation kinetics adequately. We conclude hence that both fibril-end elongation and secondary nucleation are affected by Bri2 BRICHOS. From the combined rate constants determined from the global fit and the elongation rate from the seeded aggregation kinetics, the rate of secondary nucleation can be estimated at a constant Bri2 BRICHOS concentration (Fig. 4d). This comparison makes evident that the dimer is not only most efficient in suppressing elongation events but also effectively prevents secondary nucleation events catalyzed on the fibril surface.

**Visualization of Bri2 BRICHOS interaction with Aβ42 fibrils.** The kinetic modulation of secondary nucleation and elongation events implicates an association of Bri2 BRICHOS to the fibril surfaces and fibril ends. To visualize this interaction we used anti-Bri2 BRICHOS immunogold-staining, which enabled us to localize Bri2 BRICHOS on TEM images. These images reveal Bri2 BRICHOS on the Aβ42 fibril surfaces and close to fibril-ends as well (Supplementary Fig. 8d–g), which support the kinetic analysis and provide evidence for an interaction of Bri2 BRICHOS with both the fibril surface and fibril ends. Furthermore, in the presence of rh Bri2 BRICHOS monomer, but not the dimer, Aβ42 small aggregates surrounded by rh Bri2 BRICHOS were observed (Supplementary Fig. 8h, i). This suggests the Bri2 BRICHOS monomers specially bind to non-fibrillar Aβ42 assemblies.

**Aβ42 toxicity prevented by different rh Bri2 BRICHOS species.** We next tested how the effects on Aβ42 aggregation of the different Bri2 BRICHOS species translated into effects on Aβ42 neurotoxicity. We recorded γ oscillations in mouse hippocampal slices after acute exposure to Aβ42 and different rh Bri2 BRICHOS species. In general, the characteristics of γ oscillations are important functional biomarkers for brain disorders that involve cognitive decline, since this brain rhythm plays a central role in higher processes, such as learning, memory and cognition[41, 42]. Clinical data shows that the cognitive decline observed in AD patients goes hand-in-hand with a decrease of γ oscillations[43], and it is assumed that the reduction of these network rhythms underlies the negative effects on learning, memory, perception and cognition typical for AD. γ oscillations were induced in horizontal hippocampal slices from C57BL/6 mice by superfusing slices with 100 nM kainate (KA) as a control (Fig. 5 and Supplementary Table 4). Pre-incubation of hippocampal slices with 50 nM Aβ42 for 15 min severely reduced the power of γ oscillations generated by subsequent KA application (Fig. 5, KA vs. Aβ42: $p < 0.0001$).

We found that the individual incubation with all rh Bri2 BRICHOS species reduces Aβ42-induced toxicity (Fig. 5), while control experiments showed that pre-incubation with 50 nM of any rh Bri2 BRICHOS species in the absence of Aβ42 did not significantly alter the power of γ oscillations generated by subsequent KA application (KA vs. rh Bri2 BRICHOS oligomer, $p = 0.925$, dimer $p = 0.775$, or monomer, $p = 0.808$).

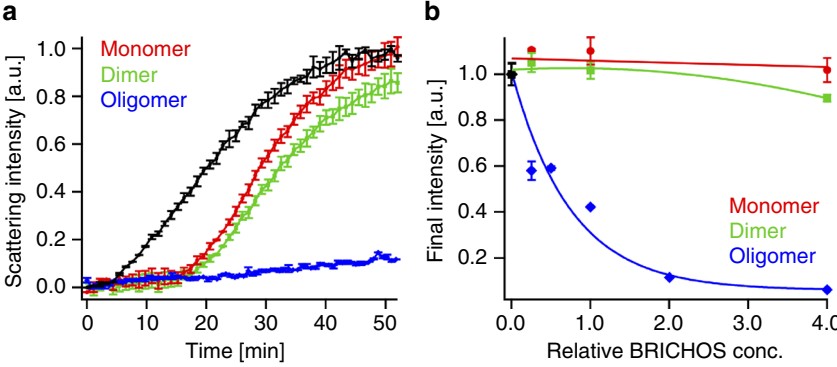

**Fig. 6** Ability of different rh Bri2 BRICHOS species to suppress citrate synthase (CS) non-fibrillar aggregation. **a** Kinetics of aggregation of 600 nM citrate synthase (CS) at 45 °C alone (black), in the presence of 1.2 μM rh Bri2 BRICHOS oligomer (blue), 2.4 μM dimer (green) or 2.4 μM monomer (red). **b** Effects of rh Bri2 BRICHOS oligomers (blue), dimers (green) and monomers (red) on CS aggregation at different molar ratios (referred to monomeric subunits) of BRICHOS:CS. The data are presented as means ± standard deviations of 3–4 replicates of experiments that have been repeated at least five times with qualitatively similar results

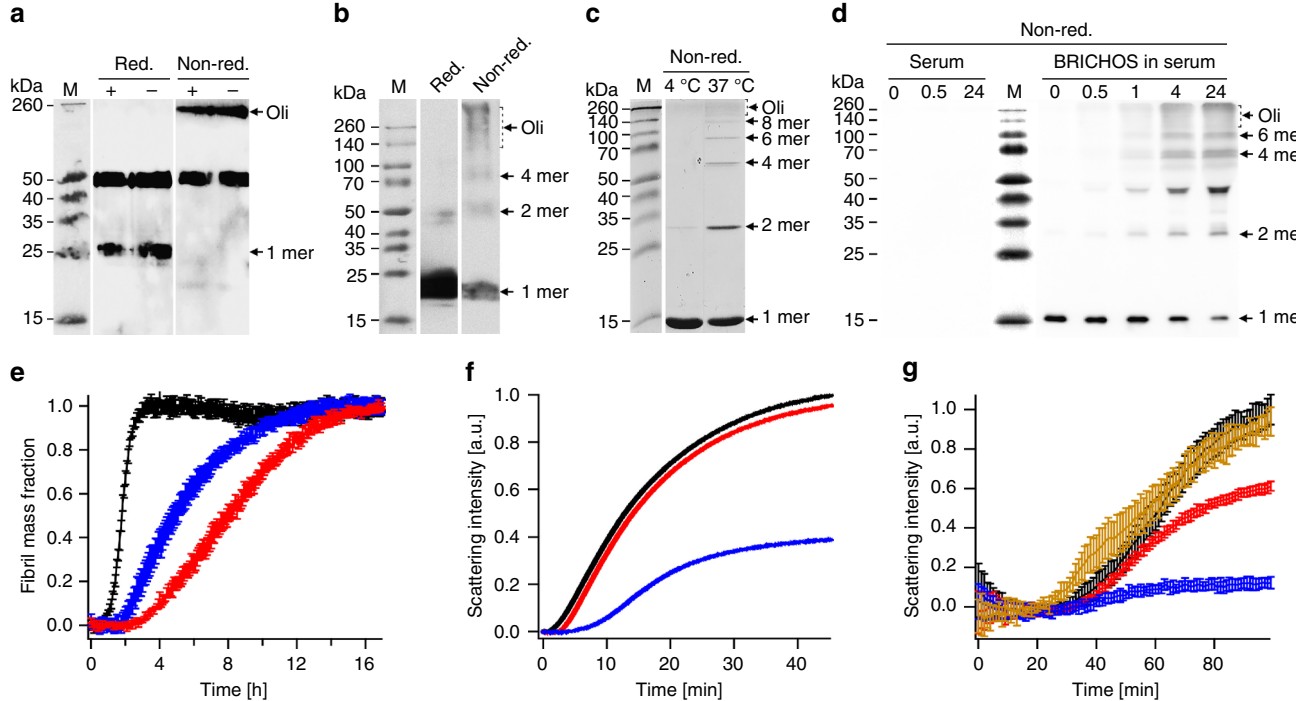

**Fig. 7** Quaternary structures and activities of Bri2 BRICHOS secreted from mammalian cells and in serum. **a** Conditioned culture medium from SH-SY5Y cells was immunoprecipitated for Bri2 BRICHOS and analyzed by SDS-PAGE and Western blot under reducing or non-reducing conditions, and in the presence (+) and absence (−) of *N*-Ethylmaleimide (NEM). The band at 50 kDa is found after the same treatment of non-conditioned medium. **b** Culture medium from HEK293 cells transfected with Bri2 BRICHOS was immunoprecipitated for Bri2 BRICHOS and analyzed by SDS-PAGE and Western blot under reducing or non-reducing conditions. **c** Monomeric rh Bri2 BRICHOS incubated at 4 °C or 37 °C overnight and analyzed by SDS-PAGE under non-reducing conditions. SDS-PAGE of the same samples under reducing conditions is shown in Supplementary Fig. 9c. **d** Monomeric rh Bri2 BRICHOS incubated in mouse serum at 37 °C, or serum only, analyzed by Western blotting under non-reducing conditions at different time points (in hours) as indicated above each lane. Western blotting of the same samples under reducing conditions is shown in Supplementary Fig. 9d. **e** Aggregation kinetics of 3 μM Aβ42 alone (black) or with 50% molar ratio of rh Bri2 BRICHOS monomer incubated at 4 °C (red) or at 37 °C overnight (blue). **f** Aggregation kinetics of 600 nM CS at 45 °C alone (black), in the presence of 1.2 μM rh Bri2 BRICHOS monomer incubated at 4 °C (red) or at 37 °C overnight (blue). **g** Aggregation kinetics of 600 nM CS at 45 °C in the presence of non-incubated serum (black), serum incubated at 37 °C (yellow), non-incubated rh Bri2 BRICHOS/serum mixture (red), and rh Bri2 BRICHOS/serum mixture incubated for 24 h at 37 °C (blue). The data are presented as means ± standard deviations of 3–4 replicates of experiments that have been repeated three times with qualitatively similar results

The rh Bri2 BRICHOS monomer is most efficient in suppressing toxic effects in neuronal network and at a 1:1 molecular ratio it completely prevents Aβ42-induced reduction of γ oscillation (Fig. 5c and Supplementary Table 4, 50 nM Aβ42 vs. 50 nM Aβ42 + 50 nM rh Bri2 BRICHOS monomer *p* < 0.0001).

The presence of rh Bri2 BRICHOS dimers and oligomers also reduces the toxic Aβ42 effect on γ oscillation power without reaching the level of complete prevention, even for a two-fold excess and the effect levels out after 1:1 Aβ42: rh Bri2 BRICHOS ratio (Fig. 5c and Supplementary Table 4).

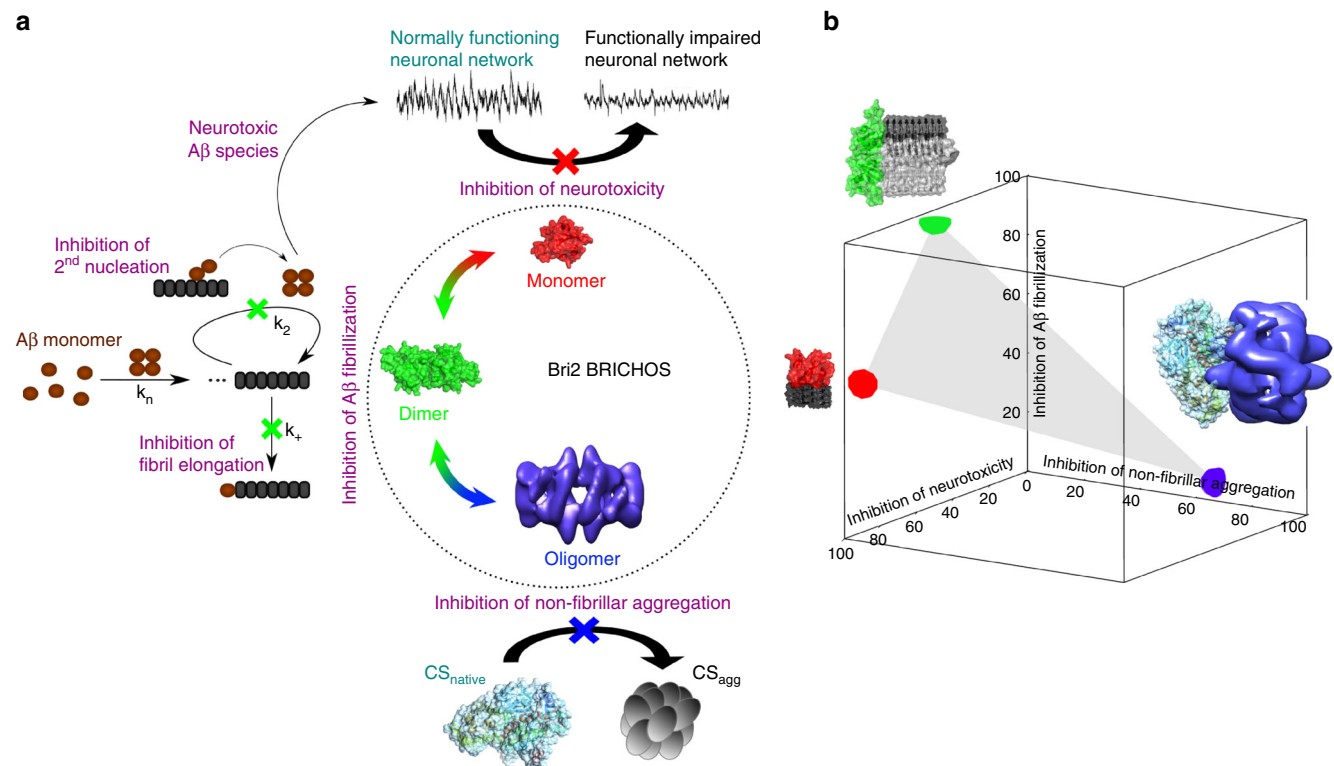

**Fig. 8** Different activities of Bri2 BRICHOS are mediated by distinct quaternary structures. **a** Bri2 BRICHOS monomers form dimers, which are able to assemble into structured high-molecular weight oligomers. Each quaternary structure most efficiently exerts one distinct function: monomers prevent Aβ42-associated neurotoxicity, dimers counteract Aβ42 fibrillization and oligomers inhibit non-fibrillar aggregation. The aggregation behavior of Aβ is governed by different microscopic nucleation events, where in particular fibril end elongation ($k_+$) and surface-catalyzed secondary nucleation ($k_2$) are inhibited by Bri2 BRICHOS dimers. **b** The efficiencies in inhibiting neurotoxicity (z-axis), Aβ42 fibrillization (y-axis) and non-fibrillar aggregation (x-axis), derived from data in Figs. 4–6, are plotted for each quaternary structure species. The diversity in functions fits with molecular sizes of the respective Bri2 BRICHOS quaternary structure compared with the sizes of their respective preferred substrate. A Bri2 BRICHOS dimer (surface area ~ 2570 Å$^2$, evaluated by juxtaposing two Bri2 BRICHOS subunit models[26], built from the proSP-C BRICHOS crystal structure (PDB 2YAD)) matches well the cross-sectional area (~ 2250 Å$^2$) of Aβ42 fibrils (PDB 5KK3) that consist of two β-structured Aβ molecules per fibril layer. A Bri2 BRICHOS monomer, which prevents Aβ toxicity to neuronal networks, fits well in size with single layer β-structured Aβ molecules that may build up oligomers. A high-molecular weight Bri2 BRICHOS oligomer (Fig. 2) that efficiently suppresses thermo-induced aggregation of CS provides a large binding surface that is suited to capture native and/or partially denatured CS (illustrated by the structure of porcine heart CS, PDB 3ENJ)

**Rh Bri2 BRICHOS activities against non-fibrillar aggregation.** Unresolved rh Bri2 BRICHOS, i.e., a mixture of different quaternary structures, suppresses aggregation of thermally destabilized citrate synthase (CS) and other substrate proteins, but it has only small effects on refolding capacity[28]. This raises the question whether the ability to suppress non-fibrillar aggregation is specifically associated with any of the now resolved different quaternary structures of rh Bri2 BRICHOS. By comparing the different species of rh Bri2 BRICHOS, we found that the dimers and monomers are quite inefficient in suppressing thermo-induced aggregation of CS; even at a four-fold molar excess to CS, the reduction in aggregation is far from being substantial (Fig. 6). In sharp contrast, however, the oligomers efficiently reduce aggregation of 600 nM CS at 45 °C already at sub-stoichiometric concentrations and completely prevent aggregation at a 2:1 molar ratio to CS (Fig. 6), indicating that the activity against non-fibrillar protein aggregation is specifically associated with Bri2 BRICHOS oligomers.

**Bri2 quaternary structures from mammalian cells and in serum.** In order to study what quaternary structure species of Bri2 BRICHOS that are found in mammalian cells we used SH-SY5Y and HEK293 cells. Immunoprecipitation and Western blot

of conditioned medium from non-transfected SH-SY5Y cells using an anti Bri2 BRICHOS antibody, revealed Bri2 BRICHOS monomer around 25 kDa under reducing conditions, and high molecular weight oligomers under non-reducing conditions, also when the thiol-blocking agent N-ethylmaleimide (NEM) is present during sample preparation (Fig. 7a). This suggests that the BRICHOS domain that is cleaved out from endogenous Bri2 expressed and secreted from SH-SY5Y cells, mainly forms larger disulfide-linked oligomers. We also recombinantly expressed Bri2 BRICHOS domain directed to the secretory pathway in HEK293 cells, and monomers, dimers, tetramers and oligomers were detected in the medium (Fig. 7b) and in cell lysates (Supplementary Fig. 9a). PNGase treatment confirms that rh Bri2 BRICHOS expressed on its own, i.e., without the rest of the Bri2 protein, gets glycosylated (Supplementary Fig. 9b), like the full-length protein[44], indicating that glycosylation of the BRICHOS domain does not affect formation of quaternary structure species.

In order to gain insights into the oligomerization mechanism of rh Bri2 BRICHOS, monomeric species from E. coli were isolated and incubated under different conditions. The monomeric rh Bri2 BRICHOS converts to disulfide-dependent dimers and even-numbered oligomers when incubated at 37 °C over night in a phosphate buffer pH 8.0, but not at 4 °C (Fig. 7c and Supplementary Fig. 9c). We also tried to mimic a physiological

environment, and incubated monomeric rh Bri2 BRICHOS in the presence of mouse serum at 37 °C, which similarly resulted in formation of disulfide-dependent dimers, tetramers, hexamers and oligomers within 0.5–1 h (Fig. 7d and Supplementary Fig. 9d). This indicates that Bri2 BRICHOS monomers can form dimers, which generate Bri2 BRICHOS oligomers with even number of subunits, and that different quaternary structure species coexist under physiological conditions. The relative occurrence of oligomers may be underestimated in these experiments since SDS-PAGE allows detection only of covalently linked oligomers.

Finally, we determined how conversion from rh Bri2 BRICHOS monomers to dimers and oligomers in buffer or serum affect the abilities to reduce CS aggregation and Aβ42 fibril formation, respectively. The rh Bri2 BRICHOS monomers, present after incubation in phosphate buffer at 4 °C, delayed Aβ42 fibrillization in the same manner as non-incubated monomers. In contrast, incubation at 37 °C overnight, which result in formation of dimers and oligomers (Fig. 7c), gives decreased anti Aβ42 fibrillization ability (Fig. 7e). The effects on the ability to inhibit CS aggregation, however, are just the opposite; after incubation at 37 °C the potency to prevent CS aggregation is increased (Fig. 7f). Likewise, the incubation of rh Bri2 BRICHOS monomers in serum, which also results in formation of oligomers (Fig. 7d), gives increased ability to prevent CS aggregation (Fig. 7g). As a control, we incubated the trimeric proSP-C BRICHOS[28, 32] at 37 °C overnight and found neither detectable formation of disulfide linked oligomers nor effects on the ability to reduce Aβ42 fibril formation (Supplementary Fig. 9e, f).

## Discussion

In the present work, we show that one single protein, the Bri2 BRICHOS domain, possesses three distinct activities: it inhibits Aβ42 fibril formation, Aβ-induced neurotoxicity and non-fibrillar protein aggregation, and these activities are associated with different quaternary structure species. The Bri2 BRICHOS monomers are able to spontaneously convert into dimers and subsequently assemble into high-molecular-weight oligomers with dimeric subunits. Notably, this quaternary structure diversity is found for recombinant Bri2 BRICHOS from both prokaryotic and eukaryotic hosts as well as for endogenous Bri2 BRICHOS secreted from SH-SY5Y cells. Moreover, isolated monomeric rh Bri2 BRICHOS converts into different quaternary structures after incubation under physiological-like conditions, which is accompanied with altered chaperone function.

The importance of protein multimerization has been recognized in relation Aβ generation, in the context of γ-secretase mediated cleavage of AβPP[45]. Our results suggest that multimerization of Bri2 BRICHOS regulates its chaperone activity and thereby its ability to affect Aβ fibrillization and neurotoxicity. Recombinant Bri2 BRICHOS high-molecular-weight oligomers efficiently inhibit non-fibrillar aggregation of thermo-denatured CS while both monomers and dimers are basically inactive and do not show any enhanced inhibitory effects (Fig. 6). In contrast, Bri2 BRICHOS dimers most efficiently suppress Aβ42 fibrillization, in particular elongation and secondary nucleation events. Also the monomeric, and to a lesser extent the oligomeric, forms retard Aβ42 self-assembly, while the dimeric structure appears to be most suited to prevent surface-catalyzed nucleation and fibril-end elongation (Figs. 3 and 4). It is crucial to find potential AD drug candidates not only by their ability to non-specifically suppress amyloid formation, but rather by their efficiency in suppressing high levels of neurotoxic oligomers. The Bri2 BRICHOS monomers are most efficacious in reducing Aβ42-induced toxicity in neuronal networks and completely prevent toxic effects

at a 1:1 Aβ42:BRICHOS molar ratio. The dimer and oligomer can also prevent Aβ42-induced damage to the neuronal network, yet their effects are limited and do not reach 100% prevention even at a higher 1:2 Aβ42:BRICHOS molar ratio. The lack of correlation between potency in inhibiting Aβ42 fibril formation and neurotoxicity supports the concept that assemblies formed during fibril formation, rather than the fibrils as such, are the main culprits in mediating toxicity.

We rationalized these results in a schematic model (Fig. 8). It is striking that the same chaperone domain can execute conceptually different functions, which are essential for the successful operation of the cell under proteostasis stress. Different quaternary structures are not only adopted by the Bri2 BRICHOS domain, but also other molecular chaperons can assemble into differently sized species, yet in a different manner from Bri2 BRICHOS. ProSP-C BRICHOS, phylogenetically in the same family as the Bri2 BRICHOS, is uniquely expressed in the lung epithelium, and the recombinant proSP-C BRICHOS mainly forms trimers. The trimer appears to be an inactive storage form, while the monomer is the active species against amyloid fibrillization[32, 46]. Interestingly, neither as trimer nor monomer, proSP-C BRICHOS possesses general chaperone activity and is not able to inhibit non-fibrillar aggregation[28]. This is in stark contrast to Bri2 BRICHOS, suggesting that BRICHOS domains from different proproteins have adopted diverse quaternary structures and functions during evolution.

Small heat-shock proteins (sHSP) are a large molecular chaperone family referred to as holdase chaperones and an important common feature of most sHSPs is the ability to assemble into large oligomers. These oligomeric assemblies are built up by dimeric blocks, which typically range from 12 to >32 subunits. The differently sized assemblies are mediated by non-conserved N- and C-terminal segments[47–49]. For performing efficient chaperone activity, most sHSPs require disassembling from inactive oligomers to smaller active species[50]. Also, a bias toward monomeric substructure of α-crystallins was proposed to be responsible for the aberrant chaperone behavior associated with protein deposition disease[51]. This mechanism is in sharp contrast to Bri2 BRICHOS, where large oligomers are the active species preventing non-fibrillar aggregation (Fig. 8).

Some molecular chaperones, including sHSP, also show inhibitory effects against Aβ fibrillization in vitro, e.g., Hsp40, Hsp70, Hsp90, Hsp104, αA-, and αB-crystallin[29, 52–54]. The diverse anti-amyloid active chaperones affect different microscopic nucleation events in Aβ fibril formation. DNAJB6 from the Hsp40 family inhibits predominately primary nucleation[29], proSP-C BRICHOS prevents secondary nucleation[18] and αB-crystallin modulates both elongation and secondary nucleation events[29], similarly as the comprehensive abilities of Bri2 BRICHOS, indicating that molecular chaperones have evolved specific capacities to interact with different Aβ species[29]. αB-crystallin was recently reported to use different interfaces to capture amorphous and amyloid clients, respectively. A hydrophobic edge of the central β-sandwich of αB-crystallin preferentially binds Aβ peptides, while amorphously aggregating clients are captured by the partially disordered N-terminal domain[55]. Inhibitory functions that are associated with distinct binding sites for amorphous and amyloid aggregates, as for αB-crystallin, are conceptually different to a quaternary structure dependent functional diversity, which is exemplified by Bri2 BRICHOS. In addition to sHSPs, a number of other molecular chaperones or chaperone-like proteins have been reported to inhibit amyloid formation, among them clusterin and calcium-binding protein nuclebindin-1[56, 57]. Clusterin can assemble into a variety of oligomeric forms built up by heterodimers via non-covalent association[56]. Clusterin was reported to suppress non-fibrillar aggregation induced by stress and to inhibit

Aβ fibrillization as well as to reduce Aβ-associated toxicity, and the dissociation into dimers enhances its chaperone activity[56, 58, 59]. Even though many aspects of the mechanism of action are still unclear, it appears that the functioning of clusterin's protective chaperone activities is similar to sHSPs, but separated from Bri2 BRICHOS.

In contrast to many other molecular chaperones, the oligomeric form of Bri2 BRICHOS most efficiently acts against non-fibrillar aggregation, which might be related to distinct structural arrangement and binding surface. Indeed, a change of the exposed hydrophobic surface upon oligomer formation is indicated by a modulated bis-ANS fluorescence (Supplementary Fig. 2e). Hence, larger and/or more exposed hydrophobic regions may be present in the oligomeric assemblies for substrate binding. Furthermore, the distinct symmetric structure of Bri2 BRICHOS oligomers presented in this study provides the capability to efficiently capture native and/or partly denatured CS (Fig. 8b).

The dimeric structure of Bri2 BRICHOS is superior in inhibition of Aβ42 fibril formation compared to the Bri2 BRICHOS monomer and oligomer. We found that both elongation and secondary nucleation events are inhibited most efficiently in the presence of the Bri2 BRICHOS dimer, suggesting that the dimer more comprehensively covers the fibril-ends and the fibril surface. Compared to the Bri2 BRICHOS monomers, the absolute number of dimers is only half at the same mass concentration, indicating that the dimer binds much stronger to the Aβ42 fibril-end and shields the fibril surface more efficiently from secondary nucleation. For the Bri2 BRICHOS oligomer, crucial amyloid interacting surfaces might be, at least partly, buried in the interior of the structure, thereby reducing its inhibitory effect against Aβ42 fibril formation. Moreover, the Bri2 BRICHOS oligomer is present in a much smaller absolute number than the dimer, diminishing the ability to effectively cover fibril surfaces and ends. Recently, two independent studies reported a 3D Aβ42 fibril structure with two Aβ42 molecules per fibril layer, forming a dimeric cross-β sheet entity[60, 61]. The size of the Bri2 BRICHOS dimer matches remarkably well the cross-sectional area of such a dimeric β-sheet motif, facilitating a large interaction surface and efficient binding (Fig. 8b).

The monomeric form of Bri2 BRICHOS most effectively suppresses Aβ42-induced neurotoxicity, reaching a prevention effect of γ-oscillation up to 100%, whereas the other Bri2 BRICHOS species do not yield complete prevention. Considering the present results, it appears that the small qualitative differences among the different Bri2 BRICHOS species in the $k_2/k_+$ ratio, which is related to the amount of generated Aβ42 oligomers, are dependent on the present Bri2 BRICHOS concentration (Figs. 3 and 4). This makes extrapolation of the Bri2 BRICHOS effects on Aβ42 fibrillization to neurotoxicity experiments on hippocampal slices difficult, as they were performed in a very different concentration range. It is however striking that the size of the Bri2 BRICHOS monomer well fits a single β-sheet surface (Fig. 8b). Aβ oligomers, in contrast to mature fibrils[60, 61], can feature such a monomeric β-sheet interface, providing a binding surface for monomeric Bri2 BRICHOS. Such an interaction is also indicated by immunogold-TEM staining images (Supplementary Fig. 8d–i), revealing accumulation of Bri2 BRICHOS monomers, but not dimers, around small non-fibrillar objects, putative pre-fibrillar Aβ species. Preferential binding of the Bri2 BRICHOS monomer to small sized Aβ species may thus be the origin for an enhanced protective effect against neurotoxicity.

Bri2 undergoes several proteolytic cleavages; furin-like proteinase generates a C-terminal peptide and a membrane bound mature Bri2 that is further processed by ADAM10, which sheds the BRICHOS domain into the extracellular space[62]. Recombinant Bri2 proteins expressed in HEK293 cells form dimers and oligomers via non-covalent interactions and/or disulfide bonds formed in the ER, and disulfide linked dimers were also detected in mouse brains[63]. Recently, Bri2 levels were found to increase up to 3-fold and to form more high molecular weight oligomers in AD brain compared to healthy controls[24]. This suggests that the spectrum of Bri2 BRICHOS quaternary structures and activities (Fig. 8) can be affected by physiological as well as pathological processes. Further studies in vivo are required to test these suppositions.

In conclusion, Bri2 BRICHOS has the ability to execute distinct chaperone-like activities, i.e., prevention of non-fibrillar and fibrillar aggregation as well as a neuroprotective function, via its different assembly states. Distinct chaperone-like functions from one chaperone domain provides a means to generate molecular chaperone diversity, which potentially can be implemented in treatment strategies against amyloid-related and other protein-misfolding diseases.

## Methods

**Rh Bri2 BRICHOS preparation**. A gene fragment encoding NT*-Bri2 BRICHOS (corresponding to residue 113–231 of full-length human Bri2) was cloned into a modified pET vector (primer sequences are given in Supplementary Table 5). NT* is an efficient solubility tag for recombinant protein production derived from the N-terminal domain of spider silk proteins[31]. The construct coding for the fusion protein His$_6$-NT*-thrombin cleavage site-Bri2 BRICHOS was transfected into SHuffle T7 competent *E. coli* (K12 strain) cells. The cells were incubated at 30 °C in LB medium containing 15 μg mL$^{-1}$ kanamycin, at an OD$_{600\,nm}$ around 0.9, the incubation temperature was lowered to 20 °C, and Isopropyl β-D-1-thiogalactopyranoside (IPTG) was added to 0.5 mM and the cells were incubated overnight. The cells were then harvested by 7,000 × g centrifugation at 4 °C, after which the cell pellets were resuspended in 20 mM Tris pH 8.0, and sonicated for 5 min on ice (2 s on, 2 s off, 65% of max power). The lysate was centrifuged (50,000 × g) at 4 °C for 30 min and the supernatant containing the fusion protein was then loaded on a Ni-NTA column. Unbound proteins were washed away and the fusion protein was eluted with 300 mM imidazole, and then dialyzed overnight against 20 mM Tris pH 8.0 at 4 °C. To remove the His$_6$-NT* part, the fusion proteins were cleaved with thrombin (1:1000 enzyme to substrate, w/w) at 4 °C overnight and loaded over a Ni-NTA column.

**Rh Bri2 BRICHOS species isolated by SEC**. Different rh Bri2 BRICHOS species were separated and analyzed by Superdex 200 PG, 200 GL or 75 PG columns (GE Healthcare, UK) using an ÄKTA basic 10 FPLC system (GE Healthcare, UK). Molecular mass standards aprotinin (6.5 kDa), ribonuclease (13.7 kDa), carbonic anhydrase (29 kDa), ovalbumin (44 kDa), conalbumin (75 kDa), aldolase (158 kDa) and ferritin (440 kDa) (GE Healthcare, UK) were used for calibration. For preparation of rh Bri2 BRICHOS oligomers for EM density map determination, a Superose 6 GL column (GE Healthcare, UK) was used.

**ESI-MS on rh Bri2 BRICHOS species**. Prior to ESI-MS analysis, rh Bri2 BRICHOS fractions upon SEC were exchanged into 200 mM ammonium acetate pH 7.5 using BioSpin microcentrifuge columns (BioRad, US). Final protein concentrations (referred to monomeric subunit) were ~80 μM for rh Bri2 BRICHOS oligomers and 20 μM for the monomers. Spectra were recorded on a Waters Synapt G1 mass spectrometer (Waters, Milford, MA) modified for high mass analysis. Samples were introduced into the mass spectrometer using in-house produced gold-coated borosilicate capillaries. Instrument settings were: Capillary voltage 1.5 V, sample cone voltage 30 V, extraction cone voltage 4 V. The collision voltages in the trap were step-wise increased from 10 V to 180 V in 10 V increments. The transfer voltage was 10 V. The source pressure was increased to 7 mbar. Trap gas was N$_2$ with a flow rate of 8 mL h$^{-1}$. Data analysis was performed using Waters MassLynx 4.1 software.

**Quantification of disulfide bonds**. Rh Bri2 BRICHOS monomers were prepared in native (100 mM Tris pH 8.0), denaturing (6 M guanidine hydrochloride (Gdn-HCl) in 100 mM Tris pH 8.0) or reducing buffer (denaturing buffer with 20 fold molar excess of tris(2-carboxyethyl)phosphine (TCEP) over protein concentration). Excess of TCEP was removed after 16 h reduction at room temperature by two rounds of buffer exchange in degassed denaturing buffer over a PD-10 desalting column (GE Healthcare, UK). Samples were prepared in 1 mL acryl cuvettes, containing 1 mL one of above mentioned buffer solutions, 300 nmol DTNB (Sigma-Aldrich, Germany) and 2–4 nM Bri2 BRICHOS proteins. Free thiols were quantified by monitoring the absorbance at 412 nm (UV-1800, Shimadzu). The amount of released TNB2$^-$ was calculated using a molar absorption coefficient of 14,150 M$^{-1}$ cm$^{-1}$ in Tris buffer and 13,700 M$^{-1}$ cm$^{-1}$ in Gdn-HCl.

**CD spectroscopy and bis-ANS fluorescence.** CD spectra were recorded from 260 to 190 nm at 25 °C in 1 mm path length quartz cuvettes using an Aviv 410 Spectrometer (Lakewood, NJ, USA). The wavelength step was 0.5 nm, averaging time 0.3 s, time constant 100 ms, and bandwidth 1 nm. The spectra shown are averages of three consecutive scans. Fluorescence experiments were performed with 1 μM rh Bri2 BRICHOS oligomer, dimer or monomer in 5 mM Tris pH 7.4, incubated with 2 μM bis-ANS for 10 min at 25 °C. Fluorescence measurements were performed with a SLM-Aminco AB-2 spectrofluorimeter by using thermostated cuvette holder (Thermo Spectronic, Waltham, MA, USA). The fluorescence emission spectra of bis-ANS from 420 to 600 nm were recorded after excitation at 395 nm.

**Preparation and TEM imaging of Bri2 BRICHOS oligomers.** Rh Bri2 BRICHOS oligomers were isolated in 20 mM ammonium acetate pH 8.0 with a Superose 6 GL column, and a narrow peak fraction with protein concentration of 47 μg mL$^{-1}$ was collected. Immediately, the collected fraction was stored on ice followed by grid preparation. Aliquots (4 μL) were adsorbed onto glow-discharged continuous carbon-coated copper grids (400 mesh, Analytical Standards) for 2 min. The grids were subsequently blotted with filter paper, washed with two drops of milli-Q water and negatively stained with one drop of 2% (w/v) uranyl acetate for 45 s before final blotting and air-drying. The sample was imaged using a Jeol JEM2100F field emission gun transmission electron microscope (Jeol, Japan) operating at 200 kV. Single micrographs for evaluating quality of the sample were recorded on a Tietz 4k × 4k CCD camera, TVIPS (Tietz Video and Image Processing Systems, GmbH, Gauting, Germany) at magnification of × 72,000 (pixel size 2.08) and 1.3–2.5 μm defocus. Micrographs for data collection were recorded on a DE-20 direct electron detector (Direct Electron, USA) at a magnification of × 83,000 (pixel size 1.04 Å) and 0.8–2.1 μm defocus. Images were recorded using a frame rate of 20 frames per second and 2 s exposure time (a total of 40 frames). The accumulated dose of the whole exposure was ~62 e$^-$ Å$^{-2}$. A total of 24 images were recorded.

**Processing of single particle images.** For each exposure, the comprised frames were drift corrected using the DE_process_frames-2.7.1.py script[64]. Drift corrected images were imported to EMAN2 (version 2.12) for further processing[65]. Defocus, particle separation and amplitude contrast were evaluated with e2evalimage.py. Single particles, 3000, in different orientations were selected from the images using e2boxer.py in swarm or manual mode. False positives, such as aggregated particles and stain artifacts were discarded. For each image, the contrast transfer function (CTF) parameters were estimated on boxed out regions, 208 × 208 pixels, (containing particles) using the e2ctf.py program. A reference-free 2D classification was performed using 2,718 phase-flipped particles with e2refine2d.py. 2D classes, representing different orientations were selected for the initial 3D density map generation with e2initialmodel.py. These maps were low pass filtered to 60 Å. An approximate two-fold symmetry was revealed by 2D-classification, in agreement with biochemical data. 3D refinement was performed using e2refine_easy.py applying both C2 and D2 symmetry aiming at a final resolution of 20 Å. Initial 3D-refinements were performed with pixel size of 4.152 Å after binning the data by a linear factor 4. In the second round of refinement, the data was binned to 2.076 Å per pixel. The final map from the first round of refinements was used as input in the second. The resolution was determined based on a Fourier shell correlation (FSC) value of 0.143[66], following the gold standard FSC procedure implemented in EMAN2[67]. For testing the stability of 3D-reconstructions a third round of refinement was performed with full sized data (1.038 Å per pixel).

**Rh Bri2 BRICHOS species incubation in ThT buffer.** To evaluate whether the different rh Bri2 BRICHOS species would be scrambled during the ThT assay observation period, rh Bri2 BRICHOS oligomers, dimers and monomers at concentrations (10–15 μM), that are 2–3 times higher than the ones used in the ThT assay to enable the detection of small changes in assembly states, were incubated in ThT running buffer at 37 °C. Samples were taken out at different time points, i.e., 0, 1, 4 and 24 h, and analyzed for assembly states by SDS-PAGE under reducing and non-reducing conditions.

**Aβ42 monomer preparation and ThT assay.** Recombinant Aβ(Met1–42), here referred to as Aβ42, was produced in BL21*(DE3) pLysS E. coli (B strain) cells and purified by ion exchange[18]. Briefly, the IPTG-induced cells were lysed on ice by sonication for 3 min (2 s on and 2 s off, 65% maximum amplitude), and the pellets were collected by 24,000×g centrifugation at 4 °C for 10 min. The pellets were dissolved by 8 M urea in 10 mM Tris-HCl pH 8.0, which was then diluted with 10 mM Tris–HCl pH 8.0 to 2 M urea for co-incubation (about 20 min in cold room) with DEAE cellulose (GE Healthcare, UK). The DEAE cellulose with bound proteins was washed with 10 mM Tris-HCl pH 8.0 and 10 mM Tris-HCl pH 8.0 containing 25 mM NaCl, respectively, and the recombinant Aβ42 was finally eluted by 125 mM NaCl in 10 mM Tris-HCl pH 8.0. The eluate was passed through a 30 kDa concentration filter, and the filtrate (crude Aβ42) was concentrated by a 5 kDa concentration filter. The crude Aβ42 proteins were lyophilized overnight and re-dissolved in 7 M Gdn-HCl and then injected into a Superdex 75 column (GE Healthcare, UK) for monomer isolation in 20 mM sodium phosphate pH 8.0 with 0.2 mM EDTA and 0.02% NaN3. The Aβ42 concentration was calculated by measuring the absorbance at 280 and 300 nm with an extinction coefficient of 1,424 M$^{-1}$ cm$^{-1}$ for (A$_{280}$-A$_{300}$). Purified Aβ42 monomers were aliquoted in low-binding Eppendorf tubes (Axygene). For analysis of the kinetics of amyloid fibril formation, 80 μL solution containing 3 μM Aβ42 monomer, 10 μM ThT and different concentrations of various rh Bri2 BRICHOS species were added to each well of half-area 96-well black polystyrene microplates with clear bottom and non-binding surface (Corning Glass 3881, USA), and incubated under quiescent conditions at 37 °C. The aggregation kinetics of Aβ42 monomer with different concentrations in presence of a constant concentration (0.9 μM) of various Bri2 BRICHOS species were measured in the same manner. The fluorescence was recorded using a 440 nm excitation filter and a 480 nm emission filter (FLUOStar Galaxy from BMG Labtech, Offenberg, Germany). For preparation of Aβ42 seeds, 3 μM Aβ42 monomer was incubated at 37 °C for about 20 h, and the fibrils were then sonicated in a water bath for 3 min. For analysis of Aβ42 fibril formation kinetics in the presence of seeds, 80 μL solution containing 3 μM Aβ42 monomer, 10 μM ThT, different concentrations of rh Bri2 BRICHOS species, and 0.6 μM seeds (calculated from the original Aβ42 monomer concentration) were added at 4 °C to each well in triplicate of half-area 96-well plates and incubated under quiescent conditions at 37 °C. The fluorescence was recorded as described above. The initial slope of the concave aggregation traces was determined by a linear fit to the first 30 min. Aggregation traces were normalized and averaged using 3–4 replicates for all the experiments. For Aβ42 alone, averages were performed using two different runs with 3 replicates each.

**Analysis of Aβ42 aggregation kinetics.** Aggregation traces of 3 μM Aβ42 with different concentrations of Bri2 BRICHOS species (referring to the monomeric subunits) in molar ratios between 0 to 100 % were, first, fitted to an empirical sigmoidal equation[39, 40]:

$$F = F_0 + A/\left(1 + \exp\left[r_{max}\left(\tau_{1/2} - t\right)\right]\right)$$

where $\tau_{1/2}$ is the aggregation half time, and $r_{max}$ the maximal growth rate, $A$ the amplitude and $F_0$ the base value of the curve (Fig. 3d, e). The fitting parameter $\tau_{1/2}$ yields a linear and the $r_{max}$ a mono-exponential decaying dependence on the relative Bri2 BRICHOS concentration.

Second, the aggregation kinetics was analyzed using a kinetic nucleation model that is determined by the microscopic rate constants for primary ($k_n$) and secondary nucleation ($k_2$) events as well as for fibril-end elongation ($k_+$)[17, 37]. Kinetic traces at different initial monomer concentration, $m(0)$, can be fitted globally, where the global fit parameters for primary $\lambda$ and secondary nucleation $\kappa$ are dependent on combined nucleation rates by $\lambda = \sqrt{2 \cdot k_+ k_n \cdot m(0)^{n_C}}$ and $\kappa = \sqrt{2 \cdot k_+ k_2 \cdot m(0)^{n_2+1}}$. The parameters $n_C$ and $n_2$ are the reaction orders for primary and secondary nucleation, respectively. In this model the time dependence of the fibril mass $M(t)$ is given by[17, 38]:

$$\frac{M(t)}{M(\infty)} = 1 - \left(\frac{B_+ + C_+}{B_+ + C_+ \cdot \exp(\kappa t)} \cdot \frac{B_- + C_+ \cdot \exp(\kappa t)}{B_- + C_+}\right)^{\frac{k_\infty^2}{\tilde{k}\kappa_\infty}} \cdot \exp(-k_\infty t) \quad (1)$$

where the additional coefficients are functions of $\lambda$ and $\kappa$:

$$C_\pm = \pm\lambda^2/2/\kappa^2$$
$$k_\infty = \sqrt{2\kappa^2/(n_2(n_2+1)) + 2\lambda^2/n_C}$$
$$\tilde{k}_\infty = \sqrt{k_\infty^2 - 4C_+C_-\kappa^2}$$
$$B_\pm = (k_\infty \pm \tilde{k}_\infty)/2/\kappa$$

The kinetic data at constant Aβ42 concentration with different Bri2 BRICHOS concentrations were then analyzed again applying the kinetic nucleation model. First, each kinetic trace was fitted individually with the kinetic nucleation model described by Eq. 1 (Fig. 3a–c).

Aggregation traces of Aβ42 alone exhibit a γ-exponent of $\gamma = -1.56 \pm 0.10$ (Fig. 4a), which corresponds to a model including monomer-dependent secondary nucleation in addition to fibril-end elongation, described by Eq. 1. The aggregation traces were, second, fitted globally (Supplementary Fig. 5) and the model fits well the data when the reaction orders for primary and secondary nucleation are set to $n_C = n_2 = 2$, as reported previously for Aβ42 aggregation[17]. The results from the global fit are given in Supplementary Table 2.

Also, the aggregation traces in the presence of Bri2 BRICHOS could be fitted globally with the same model as for Aβ42 alone (Supplementary Fig. 5) and the global fitting parameters are listed in Supplementary Table 2. The fitting parameter for primary nucleation, $\lambda$, could be fixed to the same value as for Aβ42 alone for all Bri2 BRICHOS species.

We also performed a global fit of the kinetic data set at constant Aβ42 concentration and different Bri2 BRICHOS concentration where the fit was constrained such that one fitting parameter was hold to a constant value across all Bri2 BRICHOS concentration, while the second parameter was the only free parameter (Supplementary Fig. 7 and Supplementary Table 3). This fitting constraint results in that only one single rate constant, i.e., $k_n$, $k_+$ or $k_2$, is the sole fitting parameter[39, 40].

**Immunogold staining of Aβ42 fibrils.** In total 5 μM Aβ42 monomer was incubated at 37 °C with rh Bri2 BRICHOS monomer and dimer, respectively, for about 15 h, and the fibrils were collected at 4 °C by centrifugation for 1 h at 22,000×*g*. The fibrils were carefully resuspended in 20 μL TBS, of which 2 μL were applied to formvar coated nickel grids, and incubated for about 5 min. Excess solution was removed with the edge of a Kleenex paper towel. Blocking was performed by incubation in 1% BSA in TBS for 30 min, after which the grids were washed 3 × 10 min by TBS. The grids were then incubated with goat anti-Bri2 BRICHOS antibody (1:200 dilution) overnight at 4 °C, and washed 3 × 10 min with TBS. Finally the grids were incubated with anti-goat IgG-gold secondary antibody (1:40 dilution) coupled to 10 nm gold particles (BBI Solutions, UK, EM.RAG10) for 2 h at room temperature, and washed 5 × 10 min with TBS. Excess solution was removed from the grid surface with the edge of a Kleenex paper towel. For staining, 2 μL of 2.5% uranyl acetate was added on each grid (kept about 20 s), and excess solution was finally removed. The grids were dried for about 20 s, and analyzed by transmission electron microscopy (TEM, Hitachi H7100 TEM operated at 75 kV).

**Electrophysiological studies with rh Bri2 BRICHOS species.** Experiments were carried out in accordance with the ethical permit granted by Norra Stockholm's Djurförsöksetiska Nämnd (dnr N45/13 to AF). C57BL/6 mice of either sex (postnatal days 14–23, supplied from Charles River, Germany) were used in the experiments. The animals were deeply anaesthetized using isofluorane before being sacrificed by decapitation.

The brain was dissected out and placed in ice-cold ACSF (artificial cerebrospinal fluid) modified for dissection. This solution contained 80 mM NaCl, 24 mM NaHCO₃, 25 mM glucose, 1.25 mM NaH₂PO₄, 1 mM ascorbic acid, 3 mM NaPyruvate, 2.5 mM KCl, 4 mM MgCl₂, 0.5 mM CaCl₂ and 75 mM sucrose. Horizontal sections (350 μm thick) of the ventral hippocampi of both hemispheres were prepared with a Leica VT1200S vibratome (Microsystems, Stockholm, Sweden) and treated under different experimental conditions (see below) without randomization and the conditions were not blinded to the experimenter. Immediately after slicing sections were transferred to a submerged incubation chamber containing standard ACSF: 124 mM NaCl, 30 mM NaHCO₃, 10 mM glucose, 1.25 mM NaH₂PO₄, 3.5 mM KCl, 1.5 mM MgCl₂ and 1.5 mM CaCl₂. The chamber was held at 34 °C for at least 20 min after dissection. It was subsequently allowed to cool to ambient room temperature (~ 22 °C) for a minimum of 40 min. Proteins (Aβ42, Bri2 BRICHOS species and combinations thereof) were added to the incubation solution for 15 min before transferring slices to the interface-style recording chamber for extracellular recordings. While incubating, slices were continuously supplied with carbogen gas (5% CO₂, 95% O₂) bubbled into the ACSF.

Recordings were carried out in hippocampal area CA3 with borosilicate glass microelectrodes, pulled to a resistance of 3–5 MΩ. Local field potentials (LFP) were recorded in an interface-type chamber (perfusion rate 4.5 mL min⁻¹) at 32 °C using microelectrodes filled with ACSF placed in stratum pyramidale. LFP γ oscillations was elicited by applying kainic acid (KA) (100 nM, Tocris) to the extracellular bath. The oscillations were allowed to stabilize for 20 min before any recordings were carried out. No Aβ42, rh Bri2 BRICHOS species and combinations thereof were present in the recording chamber either during stabilization of γ oscillations or thereafter during electrophysiological recordings. The interface chamber recording solution contained 124 mM NaCl, 30 mM NaHCO₃, 10 mM glucose, 1.25 mM NaH₂PO₄, 3.5 mM KCl, 1.5 mM MgCl₂, and 1.5 mM CaCl₂.

Interface chamber LFP recordings were performed with a 4-channel amplifier/signal conditioner M102 amplifier (Electronics lab, Faculty of Mathematics and Natural Sciences, University of Cologne, Cologne, Germany). The signals were sampled at 10 kHz, conditioned using a Hum Bug 50 Hz noise eliminator (LFP signals only; Quest Scientific, North Vancouver, BC, Canada), software low-pass filtered at 1 kHz, digitized and stored using a Digidata 1322 A and Clampex 9.6 software (Molecular Devices, CA, USA).

Power spectral density plots (from 60 s long LFP recordings) were calculated in averaged Fourier-segments of 8,192 points using Axograph X (Kagi, Berkeley, CA, USA). Oscillation power was calculated by integrating the power spectral density between 20 and 80 Hz. Data is reported as means ± standard errors of the means. For statistical analysis the Student's *t*-test (unpaired) was used. Significance levels are *$p < 0.05$; **$p < 0.01$; ***$p < 0.001$. All experiments were performed with parallel controls from the same animal/preparation.

**Bri2 BRICHOS incubation in buffer and in serum.** The procedures performed involving animal tissues were in accordance with the ethical permit granted from Stockholm Södra Djurförsöksetiska Nämnd (dnr S 6–15). Adult male mice C57BL/6 were anesthetized with a solution of ketamine/xylazine, then the thoracic cavity was carefully opened and the heart exposed. Intracardial blood collection was performed with a 0.5 mL syringe from the right atrium. From each mouse 0.5 to 1 mL blood was collected, coagulated and centrifuged at 10,000 rpm for 15 min at 4 °C to collect serum. Rh Bri2 BRICHOS monomer (5 μM) was incubated at 37 °C in 80% (v/v) serum overnight, and serum alone with same amount of PBS as control. Samples were taken out at different time points, diluted in PBS, and added to reducing or non-reducing SDS loading buffer. The samples were separated using 13.5% SDS-PAGE gels and the oligomerization was analyzed by western blotting with polyclonal goat anti-Bri2 BRICHOS antibody. Additionally, monomeric rh

Bri2 BRICHOS (26.8 μM) in buffer of 20 mM phosphate pH 8.0 containing 0.2 mM EDTA and 0.02% NaN₃ was incubated at 37 °C overnight, and the degree of oligomerization was analyzed by SDS-PAGE under reducing and non-reducing conditions, the activity against Aβ42 fibrillization was evaluated using ThT assay described above and the activity against non-fibrillar aggregation was measured as described below. As a control, 20 μM rh proSP-C BRICHOS (residues 86–197 according to the full-length sequence) was incubated in 20 mM phosphate pH 8.0 containing 0.2 mM EDTA and 0.02% NaN3 at 37 °C overnight. The oligomerization and activity against Aβ42 fibrillization was analyzed in the same manner as for rh Bri2 BRICHOS.

**CS non-fibrillar aggregation.** CS from porcine heart (Sigma-Aldrich, Germany) in 40 mM HEPES/KOH pH 7.5 was diluted to 600 nM in the same buffer and then equilibrated at 45 °C in the presence or absence of different concentrations of various rh Bri2 BRICHOS species. The aggregation kinetics were measured in triplicate using a microplate reader (FLUOStar Galaxy from BMG Labtech, Offenberg, Germany) by reading the apparent increase in absorbance at 360 nm during incubation at 45 °C under quiescent conditions. The activity against CS aggregation of rh Bri2 BRICHOS monomers incubated at 37 °C was analyzed, using an UV–Visible Spectrophotometer (300 Bio, CARY) with thermostated quartz cuvette, by reading the apparent increase in absorbance at 360 nm.

For analysis of CS aggregation in the presence of serum-incubated rh Bri2 BRICHOS monomers, the incubation aliquots were collected at 0 h and after 24 h. CS was diluted to 600 nM and mixed with 600 nM serum-incubated rh Bri2 BRICHOS monomers or corresponding volume of PBS, and analyzed in triplicate. Additional samples containing equal volumes of serum incubated without Bri2 BRICHOS for 0 or 24 h were analyzed. Five replicates per sample were pipetted in half-area 96-well black polystyrene microplates with clear bottom and nonbinding surface (Corning Glass 3881, USA), each containing 150 μL samples. The aggregation was measured in a microplate reader as described above. The data were baseline corrected for each individual measurement, averaging the replicates and subtracting the corresponding blank from each CS containing sample.

**Rh Bri2 BRICHOS from mammalian cells and deglycosylation.** Human embryonic kidney (HEK) cell derived HEK293 cells (Thermo Fisher Scientific, USA) were seeded and cultured with Expi293TM Expression Media until about 200 million viable cells were obtained. The cells were not authenticated or regularly tested for mycoplasma infection. Cells were transfected with 250 μg plasmid DNA, coding for human Bri2 BRICHOS (residues 113–231) tagged C-terminally with the antibody epitope AU1, using 1 mg mL⁻¹ PEI (Sigma Aldrich, Germany) and cultured for 12 days. The recombinant proteins were directed to the secretory pathway with the prosurfactant protein B signal peptide (residues 1–23)[68]. The conditioned media or cell lysates were loaded into a HisTrap excel 1 × 1 mL columns (Thermo Fisher Scientific, USA), then eluted with 20 mM sodium phosphate pH 7.4 containing 300 mM imidazole and analyzed by western blotting with primary polyclonal rabbit anti-AU1 antibody (1:1000 dilution, Abcam, UK, ab3401) and secondary polyclonal donkey anti-rabbit IgG/HRP antibody (1:5000 dilution, GE Healthcare, UK, NA934). Alternatively, HEK293 cells (7 × 10⁵ cells per flask) were seeded into T75 cell culture flasks and cultured with Dulbecco's modified Eagels's medium (Gibco™ DMEM, high glucose, GlutaMAX™; Thermo Fisher Scientific, USA) containing 10% FBS (Gibco™ Thermo Fisher Scientific, USA) until 90% confluence. Cells were transfected with 1.0 μg plasmid DNA, encoding rh Bri2 BRICHOS-AU1 in a pcDNA3.4-TOPO vector using 0.03 mg mL⁻¹ Lipofectamine® 2000 (Thermo Fisher Scientific, Waltham, USA). As control no plasmid was transfected. During protein expression cells were cultured in FBS free DMEM medium for 24 h. The transfected cells were incubated with *N*-Ethylmaleimid (NEM, Sigma-Aldrich, Germany) diluted in PBS pH 7.4 to a final concentration of 50 mM for 10 min at room temperature; equal amounts of PBS were used as a control. The cells were lysed in lysis buffer (100 mM Tris-HCl, 200 mM NaCl, 2 mM EDTA, 2% Triton-X), containing 50 mM NEM. The total protein concentration was determined using the Bradford protein assay (Bio-Rad, Germany). Presence of N-linked glycans of rh Bri2 BRICHOS was analyzed by incubating control and transfected HEK293 cell lysates with 5 U N-Glycosidase F (PNGase F) (Roche, Mannheim, Germany) overnight at 37 °C. Reduced and non-reduced samples (50 μg total protein/sample) were prepared and loaded on 13.5% SDS gels, and subsequently analyzed by western blotting with primary polyclonal rabbit anti-AU1 antibody (1:1000 dilution, Abcam, UK, ab3401) and secondary polyclonal donkey anti-rabbit IgG/HRP antibody (1:5000 dilution, GE Healthcare, UK, NA934), and visualized in a CCD camera (Fujifilm LAS-3000, Japan).

Cells of the human neuroblastoma derived cell line SH-SY5Y (7 × 10⁵ cells per flask) (ATCC, Sweden) were seeded into T25 flasks and cultured in DMEM/F-12 medium (Gibco™ DMEM/F-12, GlutaMAX™; Thermo Fisher Scientific, USA), containing 10% FBS until 60% confluence was reach. The cells were not authenticated or regularly tested for mycoplasma infection. The cell culture was continued for 24 h with FBS free medium. Before the medium was collected NEM, or PBS as a control, was added as described above. Prior to immunoprecipitation detached cells were removed by centrifugation from the media and pre-cleared with Protein A sepharose (GE Healthcare, UK) for 1 h at 4 °C. The supernatants were incubated with polyclonal goat anti-Bri2 113–231 antibody (1:250 dilution) for 1 h at 4 °C and samples were incubated with Protein A sepharose for 1 h at 4 °C.

After washing the sepharose three times with PBS, SDS buffer was added and the samples were heated at 96 °C for 10 min. The analysis was done by western blotting with using primary polyclonal goat anti-Bri2 113–231 antibody (1:250 dilution) and secondary polyclonal rabbit anti-goat IgG/HRP antibody (1:5000 dilution, Thermo Fisher Scientific, USA, 61-1620), and visualized using a CCD camera (Fujifilm LAS-3000, Japan).

**Data availability**. The density map of the Bri2 BRICHOS oligomer have been deposited in the Electron Microscopy Data Bank (EMDB) under the accession code EMD-3918. The other data that support the findings of this study are available from the corresponding author upon reasonable request.

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

## Acknowledgements

We are grateful to Drs. Claes Andréasson and Anna Rising for helpful comments on this manuscript. This study was supported by a KID PhD studentship grant (FRK), the Swedish Research Council (J.J., A.F., H.H.), the Center for Innovative Medicine (CIMED) (J.J., H.H.), the Swedish Alzheimer foundation (A.F.), the Swedish Brain Foundation (A.F.), the Strategic Program in Neurosciences at the Karolinska Institutet (A.F.), the Stiftelsen för Gamla Tjänarinnor (G.C., A.A., H.B.), Instruct R&D pilot project grant APPID 272 (A.A.), Loo and Hans Osterman Foundation (G.C., A.A.), Geriatric Diseases Foundation at Karolinska Institutet (G.C., A.F.), VIAA Latvia NFI/R/ 2014/023 grant (H.B.) and the InnovaBalt project at Latvian Institute of Organic Synthesis (H.B.).

## Author contributions

G.C., H.E.N. A.L., Y.A.T., S.T., L.H., F.R., M.L. performed experiments. A.A. performed the kinetic analyses. G.C., A.A, H.E.N., H.H., P.J.B.K., J.P., H.B., A.F., and J.J. analyzed the data. J.J. conceived and supervised the study. G.C., A.A., and J.J. wrote the paper. All authors discussed the results and commented on the manuscript.

## Additional information

**Competing interests:** The authors declare no competing financial interests.

