## [Peer Review File · Nature Communications]

Reviewers' comments:

Reviewer #1 (Remarks to the Author):

Chen and colleagues present an intriguing set of data in the exploding literature showing BRICHOS-related suppression of amyloid formation. In a set of careful and comprehensive experiments they show the inhibition of primary nucleation, secondary nucleation, and growth is accomplished by different aggregated species of Bri2 BRICHOS.

It is indeed refreshing to see a paper that has intelligently utilized the framework developed by Knowles, Dobson, et al in order to explore the phenomena of interest, since this in fact is a major value of a mechanism, viz. to allow one to attribute behavior viewed macroscopically to microscopic events.

The work appears sound. One concern is that, when different sized aggregates display different behavior, it is important to demonstrate that the aggregate size is stable over the time frame of the experiment. Now, were this not the case it is unlikely they would obtain the same quality of global fit, since the slowest experiments would be "contaminated", but it would be nice to have this concern of stability addressed.

It is also important to realize that their data can shed light on important structural features. While the structure of the average fibril seems to be relatively well established, it is not a given that the secondary nucleation sites are part of that structure, or instead represent some type of surface disorder. Hence the ability of the BRICHOS aggregates to affect aggregation can also shed light, or at least provide limitations, on the way in which the secondary pathway functions.

Finally, I would challenge the title of the work as representing the quaternary structure. What is really being shown is the structure of the aggregates; there is no evidence that, for example, one structure of the dimer is active, and one is not. If the dimer forms, it is inhibitory. Thus it appears to this review that it is the aggregation state that is the more pertinent.

Reviewer #2 (Remarks to the Author):

Chen et al. reported different chaperone functions of Bri2 BRICHOS protein complex for amyloid-beta peptide structure. They further identified chaperone activities corresponding to each stoichiometries of Bris2 BRICHOS. This finding is critical to study neurotoxicity by A β multimer formation mediated by Bris BRICHOS. They further analyzed their structural properties using multiple methods, and reconstructed the structure using negatively stained TEM single particle analysis. The analysis methodologies are reasonable and their analytic parameters are appropriate.

The finding of different chaperone functions according to different stoichiometries is biologically new, and has critical physiological importance in general biology, and especially to control A β neurotoxicity in Alzheimer's diseases in future. They successfully performed a single particle reconstruction of the supermolecular complex comprised of more than 20 subunits, although the resolution is limited to 17 angstrom, which is relatively good considering the difficulty in the reconstruction of such supermolecular complexes with stoichiometric flexibility. However before recommending this paper to the Nature communications, the following points should be addressed.

Major

Typical raw images of particles are necessary before figure 2a to publish the reconstruction of this important protein. For the raw images, supplemental figure is not adequate, and supplemental figure 3a is too small to recognize each particle.

Minor

As for γ -secretase related to A β production, multimeric proteins with high molecular weights is known to possess a high enzyme activity, and structurally analyzed in Ogura et al. *Biochem Biophys Res Comm* 343, 525–534, 2006. It could be interesting to cite the paper in discussion as another example of the A β regulation by quaternary protein structures.

Reviewer #3 (Remarks to the Author):

The present manuscript reports different mechanisms via Bri2 BRICHOS monomers, dimers and oligomers reducing A β fibril formation and A β toxicity, indicating novel chaperone-mediated processes. The findings provide further insight and opportunities to modulate A β -related pathology.

Closest to my field of expertise, I have a methodological question considering the in vitro electrophysiological study. It is stated that pre-incubation of hippocampal slices with 50 nM A β 42 for 15 min severely reduced the power of gamma-oscillations generated by subsequent KA application. However, in the methods it is described that gamma-oscillations elicited by kainic acid were allowed to stabilize for 20 min before any recordings were carried out. Therefore, data presented are recordings after 35 min of exposure of A β 42 +/- Bri2 BRICHOS species? This point should be clarified. The referenced for gamma-oscillations and Alzheimer's disease are misleading. Neither ref. 39 or 40 demonstrate connections between gamma oscillations and learning, memory or cognition. Referencing Singer review (1999) for the statement of "cognitive decline observed in AD patients goes hand-in-hand with a decrease of γ oscillations" is totally wrong. No matter that this is a fine review on cortical neuronal synchrony, it does not even mention Alzheimer's disease at all. Hopefully, the authors will find a more fitting references and a more comprehensive discussion on AD and gamma oscillation.

The reviewer agrees with the conclusion that Bri2 BRICHOS has the ability to execute distinct chaperone-like activities, however further studies are warranted to demonstrate its functional role in A β 42 pathology and the proposed mechanisms in A β 42 toxicity taking place in in vivo conditions.

Reviewer #4 (Remarks to the Author):

Chen et al describe three distinct oligomeric states of BRICHOS with different activities towards amyloidic and amorphous aggregates. The findings are relevant as these activities are of potential interest to target extracellular aggregation in Alzheimer. The oligomeric state-specific activity swap is also interesting from a protein-chemical point of view.

There are some concerns the authors should address:

1. What is the indication that the activity of the oligomers is of physiological relevance? background for this questions are:
 - (a) neurotoxicity is linked to fibrillar aggregates or their oligomeric precursors - why shall we care about the non-fibrillar aggregates?
 - (b) most of BRICHOS shown in the gels are the monomers fractions - are oligomers of BRICHES quantitatively relevant?

(c) what determines and controls the distribution between monomers, dimers and oligomers?

2. The authors conclude from EM data that BRICHOS binds end-on to fibrils. This is not sufficiently clear in the EM pictures, which are also not adequately annotated.

3. What is the molecular basis for the interaction of BRICHOS with fibrils and aggregates? This is not sufficiently illustrated and discussed. In particular:

(a) Are there mutants in BRICHOS shifting the equilibrium?

(b) How is oligomeration of BRICHOS organised?

Reviewers' comments:

Reviewer #1 (Remarks to the Author):

Chen and colleagues present an intriguing set of data in the exploding literature showing BRICHOS-related suppression of amyloid formation. In a set of careful and comprehensive experiments they show the inhibition of primary nucleation, secondary nucleation, and growth is accomplished by different aggregated species of Bri2 BRICHOS.

It is indeed refreshing to see a paper that has intelligently utilized the framework developed by Knowles, Dobson, et al in order to explore the phenomena of interest, since this in fact is a major value of a mechanism, viz. to allow one to attribute behavior viewed macroscopically to microscopic events.

The work appears sound. One concern is that, when different sized aggregates display different behavior, it is important to demonstrate that the aggregate size is stable over the time frame of the experiment. Now, were this not the case it is unlikely they would obtain the same quality of global fit, since the slowest experiments would be "contaminated", but it would be nice to have this concern of stability addressed.

Reply: We are grateful for the positive comments on our manuscript. We agree with the comment that the outcome of our slowest experiments – the A β 42 ThT fibril formation assay – would not be consistent if the different Bri2 BRICHOS species would be scrambled during the observation period, and we have performed additional experiments to address this. To this end, we incubated the Bri2 BRICHOS oligomers, dimers and monomer at concentrations that are 2-3 times higher than the ones used in the ThT assay (to enable detection of small changes in aggregation states), and in the ThT buffer, at 37°C and samples were analysed for aggregation state by SDS-PAGE after 0, 1, 4 and 24 hours. The results (Figure 1 below) confirm that the monomers are completely stable for 4 h, and even after 24 h only tiny amounts of dimers can be seen. The dimers likewise are stable and after 24 h incubation only minute amounts of larger oligomers can be discerned. For the oligomers no significant changes could be observed during 24h incubation. This strongly suggests that at the concentrations used for the longest experiments the different Bri2 BRICHOS species are stable. A sentence describing these results has been added on p. 7.

Figure 1. Presence of different aggregation states after various times of incubation of 10-15 μ M of Bri2 BRICHOS species at 37°C analysed by SDS-PAGE under reducing (upper panel) and non-reducing (lower panel) conditions.

It is also important to realize that their data can shed light on important structural features. While the structure of the average fibril seems to be relatively well established, it is not a given that the secondary nucleation sites are part of that structure, or instead represent some type of surface disorder. Hence the ability of the BRICHOS aggregates to affect aggregation can also shed light, or at least provide limitations, on the way in which the secondary pathway functions.

Reply: we agree and have inserted this comment and a reference to the 2015 review by Singh and Balch on page 4-5.

Finally, I would challenge the title of the work as representing the quaternary structure. What is really being shown is the structure of the aggregates; there is no evidence that, for example, one structure of the dimer is active, and one is not. If the dimer forms, it is inhibitory. Thus it appears to this reviewer that it is the aggregation state that is the more pertinent.

Reply: our use of quaternary structure included only the number of subunits but we concur with the comment and have changed the title.

Reviewer #2 (Remarks to the Author):

Chen et al. reported different chaperone functions of Bri2 BRICHOS protein complex for amyloid-beta peptide structure. They further identified chaperone activities corresponding to each stoichiometries of Bri2 BRICHOS. This finding is critical to study neurotoxicity by A β multimer formation mediated by Bri2 BRICHOS. They further analyzed their structural properties using multiple methods, and reconstructed the structure using negatively stained TEM single particle analysis. The analysis methodologies are reasonable and their analytic parameters are appropriate.

The finding of different chaperone functions according to different stoichiometries is biologically new, and has critical physiological importance in general biology, and especially to control A β neurotoxicity in Alzheimer's diseases in future. They successfully performed a single particle reconstruction of the supermolecular complex comprised of more than 20 subunits, although the resolution is limited to 17 angstrom, which is relatively good considering the difficulty in the reconstruction of such supermolecular complexes with stoichiometric flexibility. However before recommending this paper to the Nature communications, the following points should be addressed.

Major

Typical raw images of particles are necessary before figure 2a to publish the reconstruction of this important protein. For the raw images, supplemental figure is not adequate, and supplemental figure 3a is too small to recognize each particle.

Reply: we agree and a high-resolution raw image has now been inserted as Figure 2a and corresponding changes have been made in the text.

Minor

As for γ -secretase related to A β production, multimeric proteins with high molecular weights is known to possess a high enzyme activity, and structurally analyzed in Ogura et al. Biochem Biophys Res Comm 343, 525–534, 2006. It could be interesting to cite the paper in discussion as another example of the A β regulation by quaternary protein structures.

Reply: we appreciate the comment and have inserted the comment as well as the reference in the Discussion on page 13.

Reviewer #3 (Remarks to the Author):

The present manuscript reports different mechanisms via Bri2 BRICHOS monomers, dimers and oligomers reducing A β fibril formation and A β toxicity, indicating novel chaperone-mediated processes. The findings provide further insight and opportunities to modulate A β -related pathology.

Closest to my field of expertise, I have a methodological question considering the in vitro electrophysiological study. It is stated that pre-incubation of hippocampal slices with 50 nM A β 42 for 15 min severely reduced the power of gamma-oscillations generated by subsequent KA application. However, in the methods it is described that gamma-oscillations elicited by kainic acid were allowed to stabilize for 20 min before any recordings were carried out. Therefore, data presented are recordings after 35 min of exposure of A β 42 +/- Bri2 BRICHOS species? This point should be clarified.

Reply: we are sorry for the unclear formulations. A β 42 and Bri2 BRICHOS species, and combinations thereof, were present in the incubation chamber for 15 min. Then slices were transferred to an interface recording chamber where A β 42, Bri2 BRICHOS species and combinations thereof were not present. 100 nM KA was added and γ oscillations were allowed to stabilise for 20 min. Then recordings commenced. We added the following sentence to the Methods section on page 24: "No A β 42, Bri2 BRICHOS species and combinations thereof were present in the recording chamber either during stabilization of gamma oscillations or thereafter during electrophysiological recordings."

The referenced for gamma-oscillations and Alzheimer's disease are misleading. Neither ref. 39 or 40 demonstrate connections between gamma oscillations and learning, memory or cognition. Referencing Singer review (1999) for the statement of "cognitive decline observed in AD patients goes hand-in-hand with a decrease of γ oscillations" is totally wrong. No matter that this is a fine review on cortical neuronal synchrony, it does not even mention Alzheimer's disease at all. Hopefully, the authors will find a more fitting references and a more comprehensive discussion on AD and gamma oscillation.

Reply: we are grateful for pointing this out and apologize for the incorrect referencing. We have now corrected the references and inserted three new references (41-43 in the revised manuscript). Buzsaki "Rhythms of the Brain" 2006 and Yamamoto et al., Cell 2014 are now referenced to illustrate the connection between γ oscillations and higher brain functions while Ribary et al., PNAS 1991 is now cited as evidence of a publication that shows the reduction of cortical γ oscillations in AD patients vs healthy controls.

The reviewer agrees with the conclusion that Bri2 BRICHOS has the ability to execute distinct chaperone-like activities, however further studies are warranted to demonstrate its functional role in A β 42 pathology and the proposed mechanisms in A β 42 toxicity taking place in in vivo conditions.

Reply: we agree and have added a sentence at the end of the Discussion (page 17) to make this clear.

Reviewer #4 (Remarks to the Author):

Chen et al describe three distinct oligomeric states of BRICHOS with different activities towards amyloidic and amorphous aggregates. The findings are relevant as these activities are of potential interest to target extracellular aggregation in Alzheimer. The oligomeric state-specific activity swap is also interesting from a protein-chemical point of view.

There are some concerns the authors should address:

1. What is the indication that the activity of the oligomers is of physiological relevance? background for this questions are:

(a) neurotoxicity is linked to fibrillar aggregates or their oligomeric precursors - why shall we care about the non-fibrillar aggregates?

Reply: our main message here is the observation that one domain can execute different chaperone functions by assembling into different species. The effects of Bri2 BRICHOS on non-fibrillar aggregation may not be directly relevant for neurotoxicity, but may suggest that increasing the relative occurrence of monomers and dimers by dissociating oligomers can be a means to indirectly affect neurotoxicity. We have added a short description of the distinction between non-fibrillar vs fibrillar aggregation in the Introduction on p. 3.

(b) most of BRICHOS shown in the gels are the monomers fractions - are oligomers of BIRCHES quantitatively relevant?

Reply: the relative occurrence of oligomers may be underestimated since SDS-PAGE allows detection only of covalently linked oligomers. We now point this out on page 12.

(c) what determines and controls the distribution between monomers, dimers and oligomers?

Reply: this is a good question and we are currently studying the importance of variables such as ionic strength, pH, redox status, metal ion concentrations, and protein interactions on the distribution between different Bri2 BRICHOS species. At the present stage, however, we have no

further information in addition to what is already shown in figure 7 and Supplementary figure 8 on the effects of serum, eukaryotic expression and glycosylation.

2. The authors conclude from EM data that BRICHOS binds end-on to fibrils. This is not sufficiently clear in the EM pictures, which are also not adequately annotated.

Reply: we apologize for this and have now added two figures (Supplementary figure 7 f and g), annotated the figures and arrows have been inserted in Supplementary figure 7 to point out fibril ends covered by Bri2 BRICHOS.

3. What is the molecular basis for the interaction of BRICHOS with fibrils and aggregates? This is not sufficiently illustrated and discussed. In particular:

- (a) Are there mutants in BRICHOS shifting the equilibrium?
- (b) How is oligomeration of BRICHOS organised?

Reply: we agree with the reviewer that these questions are important. At the present stage we are working both on site-specific mutants of Bri2 BRICHOS, as well as other BRICHOS domains, with the ambition to better understand what part(s) of the subunit that mediate inter-domain contacts, and to obtain high-resolution structural data on low n as well as high n oligomers. We hope that the results of these studies will be the topics of future papers.